# Activation-triggered subunit exchange between CaMKII holoenzymes facilitates the spread of kinase activity

Margaret Stratton[1,2,3†], Il-Hyung Lee[2,3,4†], Moitrayee Bhattacharyya[1,2,3], Sune M Christensen[2,3,4], Luke H Chao[1,2,3], Howard Schulman[5], Jay T Groves[2,3,6,7]*, John Kuriyan[1,2,3,4,7]*

[1]Department of Molecular and Cell Biology, Berkeley, Berkeley, United States; [2]California Institute for Quantitative Biosciences, (QB3), University of California, Berkeley, Berkeley, United States; [3]Howard Hughes Medical Institute, University of California, Berkeley, Berkeley, United States; [4]Department of Chemistry, University of California, Berkeley, Berkeley, United States; [5]Allosteros Therapeutics, Sunnyvale, United States; [6]Materials Sciences Division, Lawrence Berkeley National Laboratory, Berkeley, United States; [7]Physical Biosciences Division, Lawrence Berkeley National Laboratory, Berkeley, United States

**Abstract** The activation of the dodecameric $Ca^{2+}$/calmodulin dependent kinase II (CaMKII) holoenzyme is critical for memory formation. We now report that CaMKII has a remarkable property, which is that activation of the holoenzyme triggers the exchange of subunits between holoenzymes, including unactivated ones, enabling the calcium-independent phosphorylation of new subunits. We show, using a single-molecule TIRF microscopy technique, that the exchange process is triggered by the activation of CaMKII, and that exchange is modulated by phosphorylation of two residues in the calmodulin-binding segment, Thr 305 and Thr 306. Based on these results, and on the analysis of molecular dynamics simulations, we suggest that the phosphorylated regulatory segment of CaMKII interacts with the central hub of the holoenzyme and weakens its integrity, thereby promoting exchange. Our results have implications for an earlier idea that subunit exchange in CaMKII may have relevance for information storage resulting from brief coincident stimuli during neuronal signaling.

*For correspondence: JTGroves@lbl.gov (JTG); kuriyan@berkeley.edu (JK)

†These authors contributed equally to this work

## Introduction

$Ca^{2+}$/calmodulin-dependent protein kinase II (CaMKII) plays a critical role in neurons, where it is a component of the molecular networks that lead to the strengthening of synaptic connections between co-active neurons, as exhibited in long-term potentiation (LTP) (*Lisman et al., 2002*; *Coultrap and Bayer, 2012*; *Lisman et al., 2012*). The early phases of LTP involve the activation of CaMKII and the consequent phosphorylation of downstream targets, such as the AMPA receptor and stargazin, which leads to increased ionic currents into the neuron as well the targeting of CaMKII to other proteins involved in LTP (*Lisman et al., 2002*; *Wayman et al., 2008*). The activation of CaMKII, for example, results in its recruitment to the NMDA receptor, and this NMDA:CaMKII complex may contribute to the maintenance of LTP (*Bayer et al., 2001*; *Sanhueza et al., 2011*; *Sanhueza and Lisman, 2013*). CaMKII also plays an essential role in modulating cardiac pacemaking and excitation-contraction coupling. Several heart diseases are associated with the hyper-activation of CaMKII (*Backs et al., 2009*; *Chelu et al., 2009*; *Neef et al., 2010*; *Rokita and Anderson, 2012*).

CaMKII is unusual among protein kinases because it is organized into a dodecameric holoenzyme (*Bennett et al., 1983*; *Woodgett et al., 1983*; *Kolodziej et al., 2000*; *Morris and Torok, 2001*;

**eLife digest** How do fleeting signals passing through the neurons of our brains become memories that can last for years or even decades? An enzyme called CaMKII is known to have an important role in the formation of memories. CaMKII adds phosphate groups to proteins—a process that is called phosphorylation—and is itself activated when calcium levels increase inside the neurons where the enzyme is found.

Individual CaMKII proteins bind together in groups of 12 to form a 'holoenzyme'. When one of the 12 subunits is activated by calcium, it can phosphorylate the other subunits in the same holoenzyme. Once this happens, the activation of CaMKII can continue after the initial rise in calcium has ceased, and this effect is thought to be involved in the formation of long-term memories.

30 years ago, Francis Crick—famous for his role in the discovery of the double helix—proposed that memory formation might involve 'memory-storage molecules' passing an activated state to unactivated molecules, and John Lisman later suggested that CaMKII could fulfil this role by swapping subunits of holoenzymes between activated and unactivated ones. Now, Stratton, Lee et al. have tested whether CaMKII can exchange subunits by using advanced microscopy to track single molecules of CaMKII labelled with fluorescent markers. This revealed that activation can cause CaMKII subunits repeatedly to mix between holoenzymes—and this only happens once a first holoenzyme has been activated.

Subunits of CaMKII join together via a central 'hub' region, but when a subunit is activated, the phosphorylated segment may interact with the hub. This weakens the connections between the subunits, thereby making it easier for subunits to exchange between holoenzymes. This process provides a mechanism by which a level of activated CaMKII can be maintained, even if some subunits become degraded and long after the disappearance of the initial activation signal.

*Rosenberg et al., 2005*; *Chao et al., 2011*; see *Stratton et al., 2013* for a recent review of CaMKII structure, shown schematically in *Figure 1A*). The holoenzyme consists of a central hub formed by the association of the C-terminal domains of CaMKII, to which the catalytic modules, consisting of the kinase domain and a regulatory segment, are attached by a variable linker (see *Figure 1B* for a schematic representation of the domains of CaMKII). There are four CaMKII genes in humans, termed α, β, γ, and δ and these generate ~40 isoforms through alternative splicing that results in variations in the linker (*Tombes et al., 2003*). The α and β isoforms are found predominantly in neurons, while the γ and δ isoforms are distributed more broadly. In this paper we focus on the human α isoform, CaMKIIα, the major isoform in neurons, which has a 30-residue linker connecting the catalytic module to the hub domain.

The regulatory segment of autoinhibited CaMKII docks within the substrate-recognition groove of the kinase domain and keeps it inactive (*Hook and Means, 2001*). An increase in calcium levels results in the binding of $Ca^{2+}$/calmodulin ($Ca^{2+}$/CaM) to the regulatory segment, thereby activating CaMKII (*Figure 1C*; *Colbran et al., 1989*; *Ikura et al., 1992*). Subsequent to activation by $Ca^{2+}$/CaM, the phosphorylation of at least three critical threonine residues within the regulatory segment modulates kinase activity and the response to calmodulin (*Figure 1C*). In contrast to most other kinases, CaMKII has no phosphorylation sites within the activation loop.

A critical phosphorylation site, Thr 286, is located in the R1 element of the regulatory segment (see *Figure 1B*; the residue numbering we use corresponds to that of a human CaMKIIα construct in which the 30-residue linker between the catalytic module and the hub domain is deleted [*Chao et al., 2011*]). The phosphorylation of Thr 286 requires one activated kinase domain to phosphorylate another kinase domain in the same holoenzyme, and trans-phosphorylation between different holoenzymes is not observed (*Hanson et al., 1994*; *Rich and Schulman, 1998*). Phosphorylation of Thr 286 prevents the R1 element from rebinding to the kinase domain, thereby conferring partial calcium-independence to the catalytic module (*Lou et al., 1986*; *Lisman et al., 2002*). This property, referred to as autonomy, prolongs the active state of CaMKII and is likely to be critical for the generation of LTP (*Saitoh and Schwartz, 1985*; *Lai et al., 1986*; *Miller and Kennedy, 1986*; *Schworer et al., 1986*; *De Koninck and Schulman, 1998*). Mutant mice with Thr 286 in CaMKIIα replaced by alanine have limited LTP generation and display impairments in learning and memory (*Giese et al., 1998*).

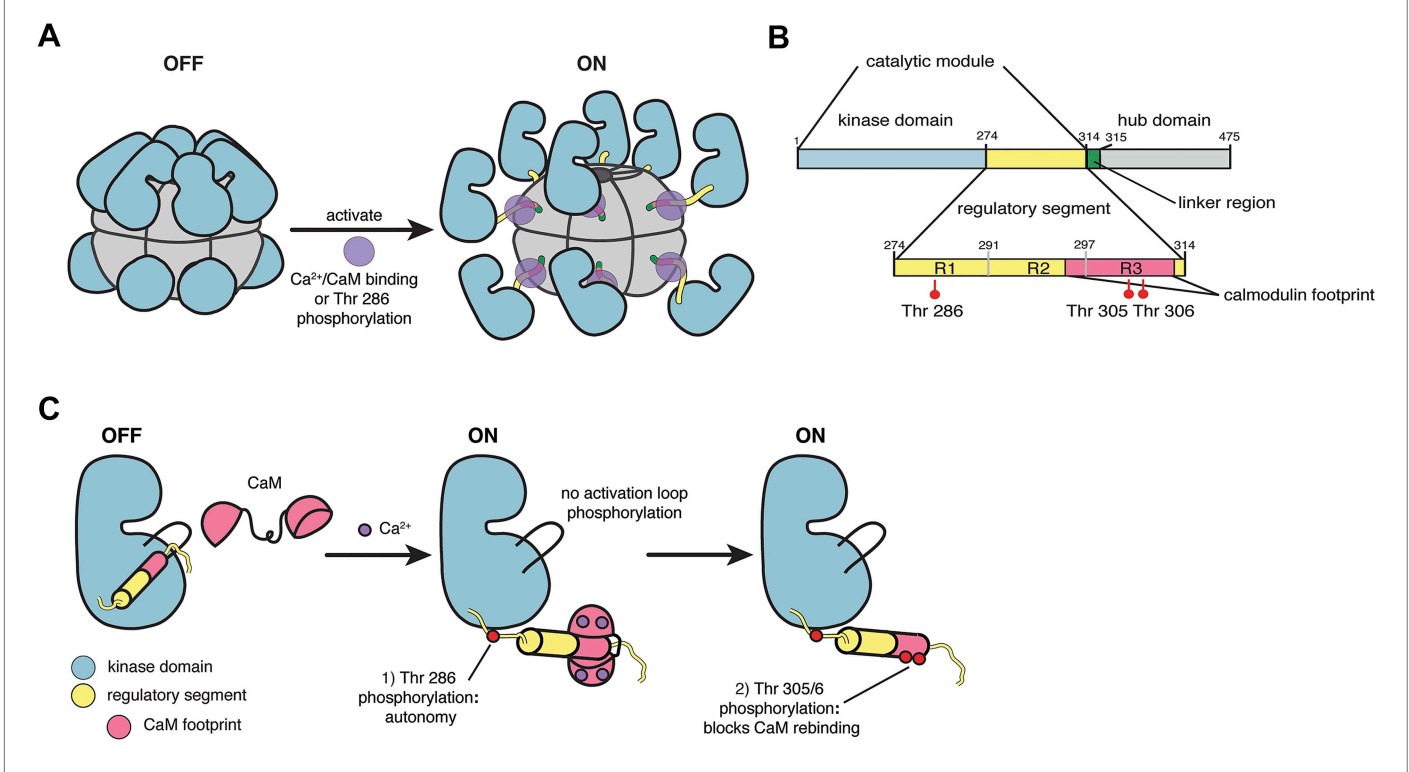

**Figure 1**. CaMKII architecture. (**A**) The architecture of a dodecameric CaMKII holoenzyme. The inactive holoenzyme is shown as a more compact configuration. Upon activation by $Ca^{2+}$/CaM, or phosphorylation of Thr 286 in the regulatory segment (purple circles), the kinase domains are extended from the hub assembly. (**B**) The domains of a CaMKII subunit. (**C**) Phosphorylation control in CaMKII. The R1 element of the regulatory segment leads into a helical R2 element that blocks the substrate binding channel of the kinase domain in the inactive form. The R3 element contains the calmodulin-recognition motif, and upon CaM binding, CaMKII is autophosphorylated at Thr 286 in the R1 element. After CaM dissociates, Thr 305 and 306 are phosphorylated if Thr 286 is already phosphorylated.

Two other key sites of phosphorylation, Thr 305 and Thr 306, are located within the calmodulin-binding portion of the regulatory segment, in the R3 element. The functional significance of these phosphorylation sites is less well understood but, as for Thr 286, studies using knock-in mice have emphasized their importance in learning and memory (*Elgersma et al., 2002*). Phosphorylation of either residue weakens the affinity of CaMKII for $Ca^{2+}$/CaM (*Hanson and Schulman, 1992*; *Colbran, 1993*).

Using in vitro experimentation, we now report that activation of the CaMKII holoenzyme enables CaMKII subunits to exchange between different holoenzymes and to spread their state of activation in the process. The possibility that memory storage in the brain might rely on the exchange of subunits between autonomous (calcium-independent) forms and unactivated forms of CaMKII had been proposed earlier by John Lisman, following an initial conjecture by Francis Crick about memory-storage molecules that have the ability to instruct newly synthesized unactivated molecules of earlier activation events (*Crick, 1984*; *Lisman, 1994*). Our results provide a mechanism for maintaining a pool of active CaMKII despite protein turnover, a process that could potentially be important for long-term information storage in the brain.

## Results and discussion

The experiments on CaMKII that we report here were prompted by a curious observation concerning the stoichiometry of subunits in CaMKII, which was that samples derived from the same CaMKII holoenzyme preparation showed both six-fold and seven-fold symmetry when visualized by electron microscopy and X-ray crystallography (*Rosenberg et al., 2006*). Crystal structures of isolated hub domain assemblies of CaMKII revealed both tetradecameric and dodecameric forms, and spectroscopic studies provided

evidence for variability in the number of subunits in intact holoenzymes (*Hoelz et al., 2003*; *Rosenberg et al., 2006*; *Rellos et al., 2010*; *Nguyen et al., 2012*). This variability in subunit stoichiometry raised the possibility that the CaMKII holoenzyme might dissociate and reassemble, allowing subunits to exchange in the process. To study this, we analyzed CaMKII subunit exchange by using a single-molecule technique to visualize fluorescently labeled holoenzymes directly. We find that although unactivated CaMKII holoenzymes do not display appreciable subunit exchange, activation by $Ca^{2+}$/CaM leads to significant subunit exchange.

All the experiments described here are carried out using the full-length human α isoform of CaMKII (CaMKIIα) and mutants thereof. Proteins are prepared using bacterial expression, as described previously (*Chao et al., 2010*, see 'Materials and methods'). Gel filtration results are consistent with the purified protein being a dodecameric form of CaMKII that has not been activated. Mass spectrometric analysis documented that the purified CaMKII is unphosphorylated prior to activation (data not shown).

## A single-molecule assay for monitoring subunit exchange in CaMKII

Our general experimental strategy is to label two samples of CaMKII separately with two different fluorophores, followed by mixing and analysis of whether holoenzymes containing both of the fluorophores are obtained. This type of analysis is complicated in bulk solution by the difficulty in distinguishing between aggregation of holoenzymes and true subunit exchange. Gel filtration analysis of a constitutively active mutant (CaMKII$^{T286D}$) shows no evidence for aggregation, and incubation of the wild-type holoenzyme with $Ca^{2+}$/CaM and ATP shows no changes in the UV-vis absorption spectrum (data not shown). We also used dynamic light scattering (DLS) to determine the size distribution of activated CaMKII assemblies in solution (for both wild-type, activated by $Ca^{2+}$/CaM, and a constitutively active mutant in which Thr 286 is replaced by aspartate [CaMKII$^{T286D}$] in the absence of $Ca^{2+}$/CaM). DLS measurements indicate that the addition of ATP to CaMKII$^{T286D}$, under the conditions used in our mixing experiments, results in a small population with sizes that are larger than that of a single holoenzyme (*Figure 2—figure supplement 1*). The DLS data for both wild-type CaMKII and CaMKII$^{T286D}$ demonstrate that there is no time-dependent change in the particle size distribution over the time scale of our experiments.

To rule out aggregation as a complicating factor definitively, we developed a single-molecule assay in which individual CaMKII holoenzymes are visualized directly. In this assay, two samples of CaMKII are labeled separately with the donor (Alexa 488, green fluorophore) and acceptor (Alexa 594, red fluorophore) using cysteine chemistry, and then mixed and incubated together in the presence or absence of $Ca^{2+}$/CaM and ATP. CaMKII holoenzymes are then immobilized on a glass slide such that individual holoenzyme assemblies are separated spatially from each other, and visualized by TIRF microscopy (see schematic diagram in *Figure 2A*). An exchange event is detected by the colocalization of red and green fluorophores, and potential aggregation is monitored by analysis of the distribution of fluorophore intensities.

For the single-molecule experiments, wild-type CaMKII was expressed with a C-terminal Avitag (i.e., the tag is located after the hub domain) (*Howarth and Ting, 2008*). The purified protein was biotinylated in vitro, and labeled separately with either the green or the red fluorophore. Mass spectrometric analysis showed that the labeling occurs at Cys 280, in the C-terminal lobe of the kinase domain. Labeling efficiencies ranged between 20 and 40% for the wild-type protein (see 'Materials and methods'), and this is sufficient for our analysis because of the direct visualization of holoenzymes. After mixing the two labeled species, a small amount of the CaMKII solution was removed at each time point, diluted, and then immobilized on the PEG-treated glass slides that were coated with streptavidin (*Figure 2A*). Multi-color TIRF imaging was performed and the extent of colocalization was determined using a custom particle-tracking program (see 'Materials and methods'). We score exchange events by treating all particles that have both red and green fluorophores equally. That is, our analysis does not distinguish between holoenzymes in which a single subunit has been exchanged and those in which more than one subunit has been exchanged.

The positions of CaMKII holoenzymes were determined by fitting a two-dimensional Gaussian function to particle intensities (see 'Materials and methods'). Importantly, the software filters out particles of insignificant brightness (i.e., noise) and aggregates (i.e., unusually bright particles) (*Figure 2—figure supplement 2*). On average, 6–20% of particles were eliminated from the analysis. Since samples from different time points are from the same reaction, the distribution of fluorescence intensities of each sample should not change over time in the absence of aggregation. Histograms of

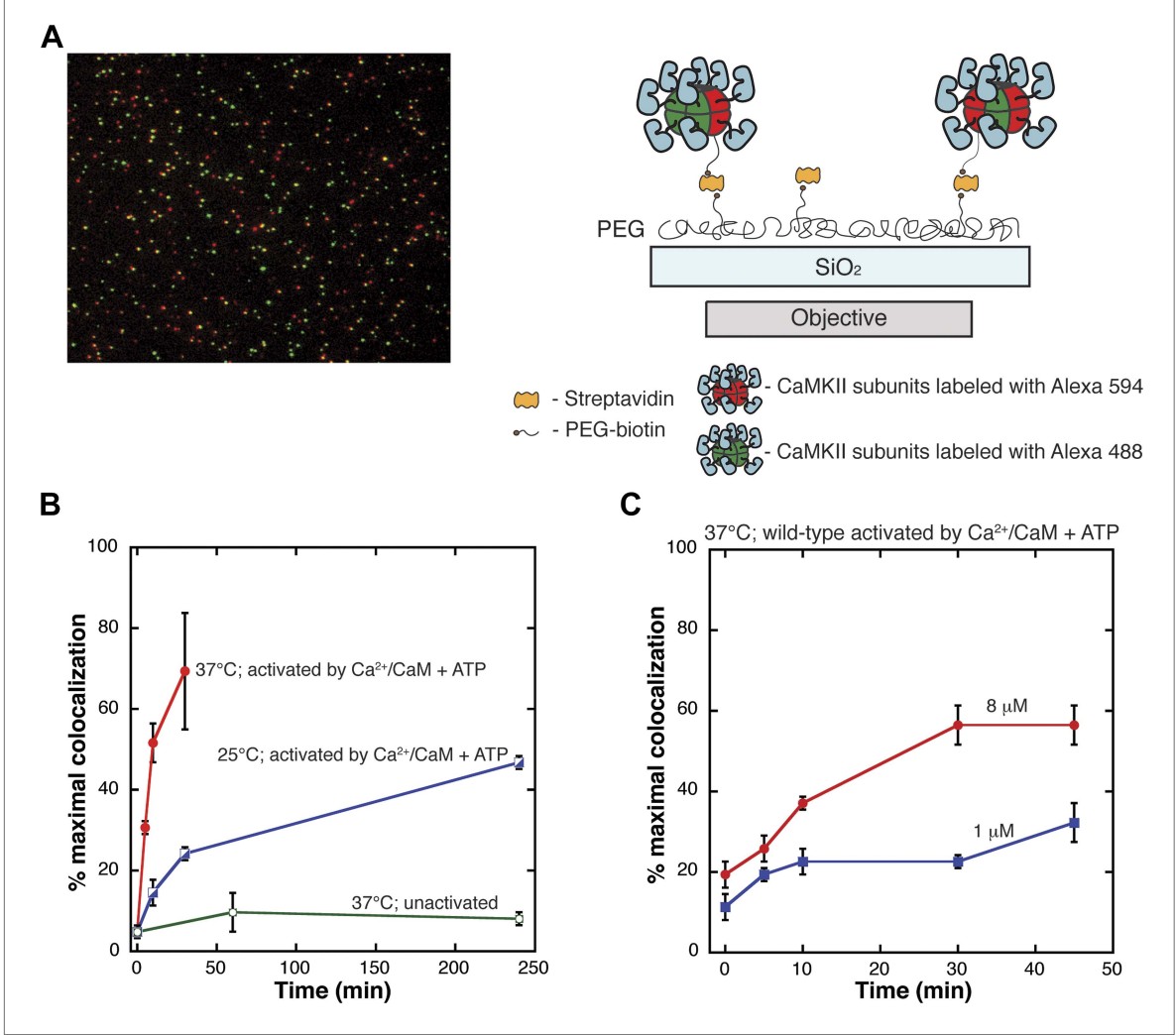

**Figure 2**. Single-molecule assay for subunit exchange reveals activation-dependent subunit exchange. (**A**) A representative single-molecule TIRF image, with red and green channels overlaid (left). For analysis, CaMKII holoenzymes are immobilized on glass slides via biotin/streptavidin interactions (right). (**B**) The rate of increase in colocalization is significantly faster at 37°C (red) compared to 25°C (blue) when $Ca^{2+}$/CaM and ATP are added. At 37°C, the unactivated sample (i.e., with no addition of $Ca^{2+}$/CaM and ATP) shows only a low level of exchange even at long time points (green). (**C**) Under activating conditions, decreasing the concentration of CaMKII from 8 µM (red) to 1 µM (blue) results in reduction of the rate of colocalization.

The following figure supplements are available for figure 2:

**Figure supplement 1**. Dynamic light scattering measurements on CaMKII.

**Figure supplement 2**. Custom particle-tracking program.

**Figure supplement 3**. Intensity distribution analysis of single-molecule images.

**Figure supplement 4**. Comparison of activation methods and subunit exchange.

**Figure supplement 5**. FRET mixing experiments corroborate single-molecule results.

the fluorescence intensities were created for each image, after processing in the particle-tracking program, and monitored to check for any population of extra-bright particles (most likely aggregates). The distribution of fluorescence intensities did not change significantly over the time course of the subunit exchange reaction, indicating that the colocalization we report is not due to aggregation,

which would cause the distribution to shift towards values of greater intensity (*Figure 2—figure supplement 3*). This does not, however, rule out that aggregation or self-association of CaMKII holoenzymes may be necessary as an intermediate step for subunit exchange to occur, since such self-association might be reversed by dilution prior to the single-molecule analysis.

## Activation by Ca²⁺/CaM and ATP triggers subunit exchange in CaMKII

We compared the results of mixing experiments in which CaMKII had either not been activated or had been activated by Ca²⁺/CaM and ATP. Protein samples were mixed and incubated together at 37°C for varying lengths of time in the absence of Ca²⁺/CaM and ATP and analyzed using the single-molecule assay. The level of colocalization detected under these conditions is very low, even at long times (~5% at 240 min, *Figure 2B*). A dramatically different result is obtained when the samples are activated by incubation for 240 min at 25°C with Ca²⁺/CaM and ATP (~40% colocalization in about 50 min, *Figure 2B*).

The absolute levels of colocalization observed in our experiments are likely to be underestimates of the true degree of colocalization. Incomplete labeling, a population of fluorescent molecules in a dark state and errors in image processing would lower the total percentage of colocalization detected. The maximum extent of colocalization (62%) was observed using a sample of CaMKII that has been labeled with both dyes simultaneously. We report the degree of colocalization in any particular experiment as a fraction of this maximal value. The values for colocalization are variable between experiments done on different days, due to labeling efficiencies and instrumental variability (compare *Figure 2B and C*). All direct comparisons made in this paper are done using data from experiments performed on the same day.

The extent of colocalization increases markedly with CaMKII concentration and with temperature, as shown in *Figure 2B,C*. For example, activated wild-type CaMKII reaches 40% of maximal colocalization within 6 min at 37°C, but at 25°C the time taken to achieve the same degree of colocalization is ~250 min. If the concentration of CaMKII subunits is reduced from 8 µM to 1 µM, the extent of colocalization in 10 min at 37°C is reduced from ~20% of maximal to ~10% (*Figure 2C*). The concentration of CaMKII subunits in dendritic spines is estimated to be ~100 µM (*Otmakhov and Lisman, 2012*), suggesting that in this environment the timescale of subunit exchange may be substantially shorter than seen in our experiments. We have not studied subunit exchange in vitro at concentrations higher than 8 µM.

We asked whether the activation-triggered colocalization involves a dead-end conversion in which holoenzymes that have undergone one cycle of exchange are inert to further exchange. To test this, we added saturating Ca²⁺/CaM and ATP to red-labeled and green-labeled forms of the holoenzyme and incubated these activated species separately for 15 min at 25°C. We then mixed the two samples and observed robust colocalization that is similar for samples that had not been incubated separately after activation (*Figure 2—figure supplement 4*). These data indicate that once CaMKII is activated, its subunits are capable of multiple cycles of exchange.

The results of the single-molecule assay are corroborated by fluorescence resonance energy transfer (FRET) measurements in solution. The site of labeling in the wild-type protein, Cys 280, is located at the periphery of the holoenzyme, and does not give a good FRET signal. We therefore labeled the hub domain at a site near the central hole in the hub assembly (residue 335), where any two such sites in the same hexameric ring would be close enough to generate a FRET signal. We replaced surface-exposed cysteines (residues 280 and 289) in the kinase domain by serine and replaced Asp 335 in the hub domain by cysteine (D335C). Labeling at this site should bring donor–acceptor pairs within ~33 Å for adjacent subunits and ~58 Å for non-adjacent subunits within a holoenzyme (*Figure 2—figure supplement 5A*). The D335C mutant CaMKII was labeled with ~80% efficiency.

The FRET signal obtained for experiments using the D335C mutant showed a marked contrast for mixed samples that were either activated by Ca²⁺/CaM and ATP or not. There is a substantial increase in the FRET signal that occurs within minutes of mixing the two labeled species for the activated sample (*Figure 2—figure supplement 5B,C*). In the unactivated sample (i.e., in the absence of Ca²⁺/CaM and ATP), there is only a modest, slow increase in the FRET signal with time, indicating that any exchange that might occur in the unactivated sample does so at a very slow rate.

## Stabilizing the dodecameric assembly by fusing CaMKII to a hexameric protein suppresses exchange

We interpret the results of the single-molecule experiments to mean that activation results in the weakening of inter-subunit contacts, facilitating exchange. To check whether this was happening, we sought to stabilize the holoenzyme assembly. To do this, we fused CaMKII to hemolysin-coregulated

protein from *Pseudomonas aeruginosa* (Hcp1), a protein that forms hexameric rings with roughly the same diameter as the hub domain of CaMKII (CaMKII-Hcp1) (*Mougous et al., 2006*) (PDB code 1Y12). The fusion protein was generated by linking the C-terminal end of the hub domain of CaMKII to the N-terminal end of Hcp1 by a 10-residue linker with a sequence that is designed to be flexible (see 'Materials and methods'). The Avitag used to immobilize CaMKII to the glass slide was incorporated after the Hcp1 sequence. The kinase activity of the CaMKII-Hcp1 fusion was tested using a peptide substrate (syntide) and it displayed cooperative activation by $Ca^{2+}$/CaM, with an activation profile similar to that of wild-type CaMKII (*Gaertner et al., 2004*; *Rosenberg et al., 2005*, data not shown).

We carried out a mixing experiment using the CaMKII-Hcp1 fusion protein in which this construct was labeled separately with either red or green dye and the two samples were mixed and incubated at 37°C. Colocalization of the two fluorophores is only ~10% even after 1 hr, compared to ~70% for the wild-type holoenzyme (*Figure 3A*). In an analogous experiment, we labeled wild-type CaMKII with the red fluorophore (Alexa 594) and the CaMKII-Hcp1 fusion protein with the green fluorophore (Alexa 488) and measured colocalization after activation (*Figure 3A*). The level of colocalization is much below that observed with the wild-type protein in this case as well.

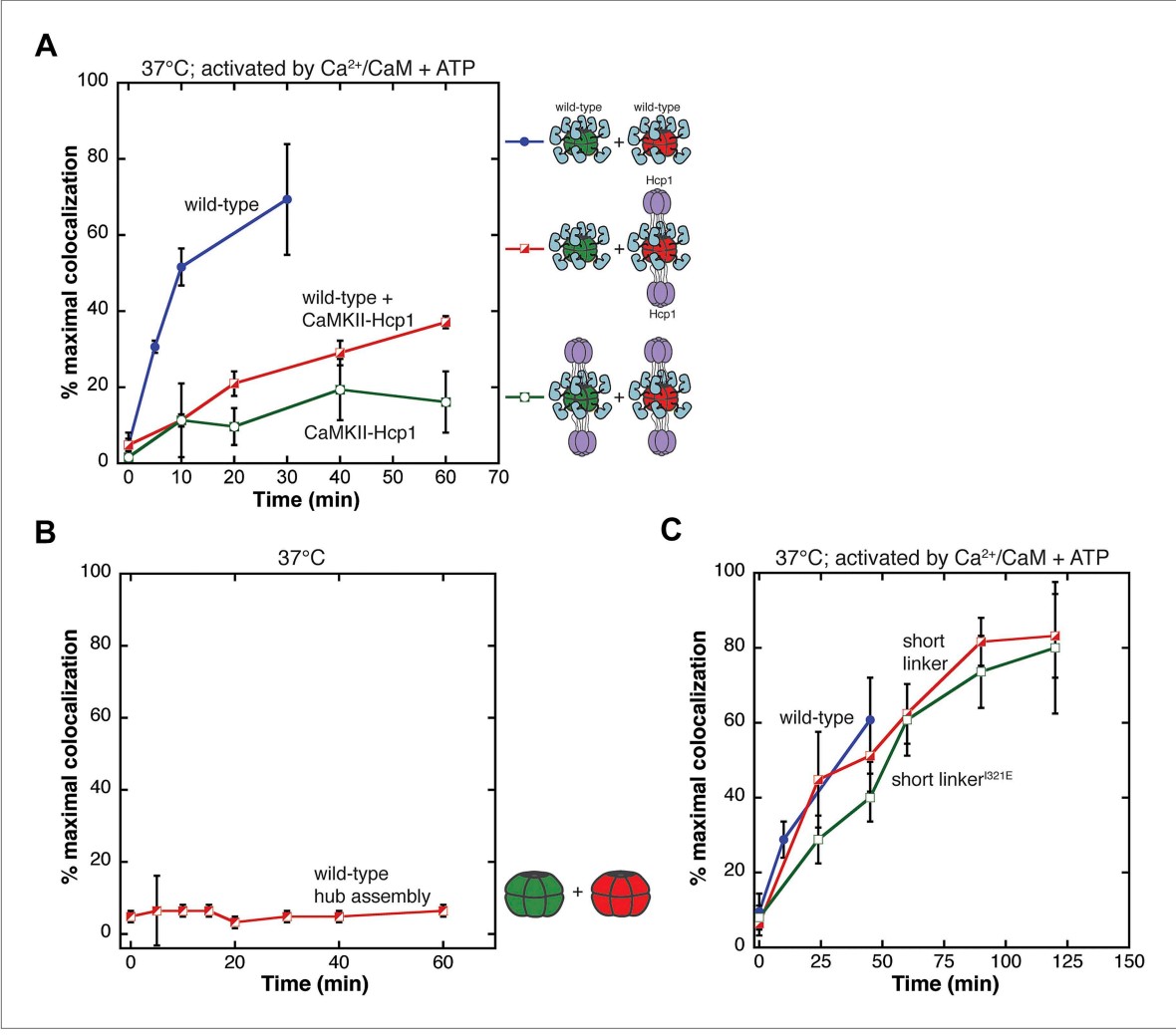

**Figure 3**. Analysis of the exchange process. (**A**) Single-molecule experiments show that fusion of CaMKII to a hexameric protein (Hcp1) slows the rate of colocalization. All samples are activated with $Ca^{2+}$/CaM and ATP and mixing is done at 37°C. Mixing activated wild-type CaMKII yields about 70% of maximal colocalization (blue). Mixing wild-type and CaMKII-Hcp1 shows a marked decrease in colocalization (red). Mixing CaMKII-Hcp1 species results in nearly no colocalization (green). (**B**) The isolated hub assembly does not result in colocalization when labeled subunits are mixed. (**C**) Deletion of the variable linker region does not affect colocalization significantly. Comparison of the short-linker construct (red), short-linker construct mutated at the hub–kinase interface (green), and wild-type CaMKII (blue) shows minimal differences in colocalization.

The strong suppression of colocalization seen with the CaMKII-Hcp1 fusion protein lends further support to the idea that CaMKII holoenzymes exchange subunits upon activation. Since fusion of Hcp1 to the hub domain is unlikely to impede the separation of a holoenzyme into two hexameric rings, these data also suggest that exchange involves some other disassembly process.

## The isolated hub domain assembly does not exchange, and the variable linker is not important for subunit exchange

The fact that activation leads to subunit exchange in the intact holoenzyme made us wonder whether the hub domain assembly might be intrinsically unstable, and that the release of stabilizing contacts between the kinase domains and the hub upon activation might allow the subunits of the hub domain to separate and exchange. To test whether the hub domain is intrinsically capable of subunit exchange, we purified the hub domain and monitored colocalization. We found that the subunits of the isolated hub domain assembly do not exchange subunits (*Figure 3B*).

These data suggest that some combination of the kinase domain, the regulatory segment or the linker connecting the regulatory segment to the hub domain must be required for the exchange process. To examine the role of the linker we carried out single-molecule experiments using a construct of CaMKIIα in which the linker is eliminated entirely. This short-linker construct is similar to the construct used to obtain the crystal structure of CaMKIIα (*Chao et al., 2011*). As shown in *Figure 3C*, the short-linker construct exhibits fluorophores colocalization with the same rate as full-length CaMKIIα when activated by $Ca^{2+}$/CaM and ATP, indicating that the linker is not required for subunit exchange.

We also wondered whether the release of interactions between the kinase domain and the hub might be the trigger for subunit exchange. In the crystal structure of the autoinhibited short-linker CaMKII holoenzyme, the kinase domains dock against the hub domains (*Chao et al., 2011*). Mutation of Ile 321 in the hub domain to glutamate disturbs this docking and results in an opening of the holoenzyme assembly (*Chao et al., 2011*). Introduction of the same mutation (I321E) in the context of the short-linker construct has no effect on the rate of colocalization (*Figure 3C*). This suggests that the trigger for subunit exchange does not involve the disruption of the interface between the kinase domain and the hub that is seen in the structure of the autoinhibited holoenzyme.

## The presence of $Ca^{2+}$/CaM is not required for exchange when CaMKII gains autonomous activity

The experiments presented so far relied on activation by $Ca^{2+}$/CaM and ATP to trigger exchange. Since the regulatory segment binds to $Ca^{2+}$/CaM, it is difficult to introduce mutations into this segment without disturbing the interaction with calmodulin. To identify the role played by $Ca^{2+}$/CaM in the exchange process, we used the constitutively activated form of the enzyme (CaMKII[T286D]) in which the regulatory segment is expected to be displaced from the kinase domain even in the absence of $Ca^{2+}$/CaM.

CaMKII[T286D] displays a vigorous degree of colocalization when ATP is added in the absence of $Ca^{2+}$/CaM, comparable to that of the wild-type enzyme in the presence of $Ca^{2+}$/CaM and ATP (*Figure 4A*). These data demonstrate that although activation of the catalytic module is necessary for colocalization, the physical presence of $Ca^{2+}$/CaM is not required.

## Phosphorylation of the calmodulin-recognition element potentiates subunit exchange

With the elimination of a direct role for $Ca^{2+}$/CaM in the exchange process our attention was now focused on the kinase domain and the regulatory segment. We were particularly interested in the part of the regulatory segment that spans the calmodulin-recognition motif (the R3 element), because this segment interacts with the hub domain in the crystal structure of the autoinhibited CaMKII holoenzyme (*Chao et al., 2011*).

Starting with the constitutively active variant (CaMKII[T286D]), we replaced residues 302 to 313 in the R3 element by an arbitrary linker sequence that is not expected to form regular secondary structure. Specifically, the sequence [302]A I L T T M L A T R N F[313] in the R3 element was replaced by [302]S T G G G T G S T G S T[313]. We monitored colocalization after mixing for this construct and found that colocalization was essentially eliminated (*Figure 4A*).

Given that the calmodulin-recognition element plays a crucial role in subunit exchange, how might activation control the ability of this segment to mediate exchange? Activation by $Ca^{2+}$/CaM releases

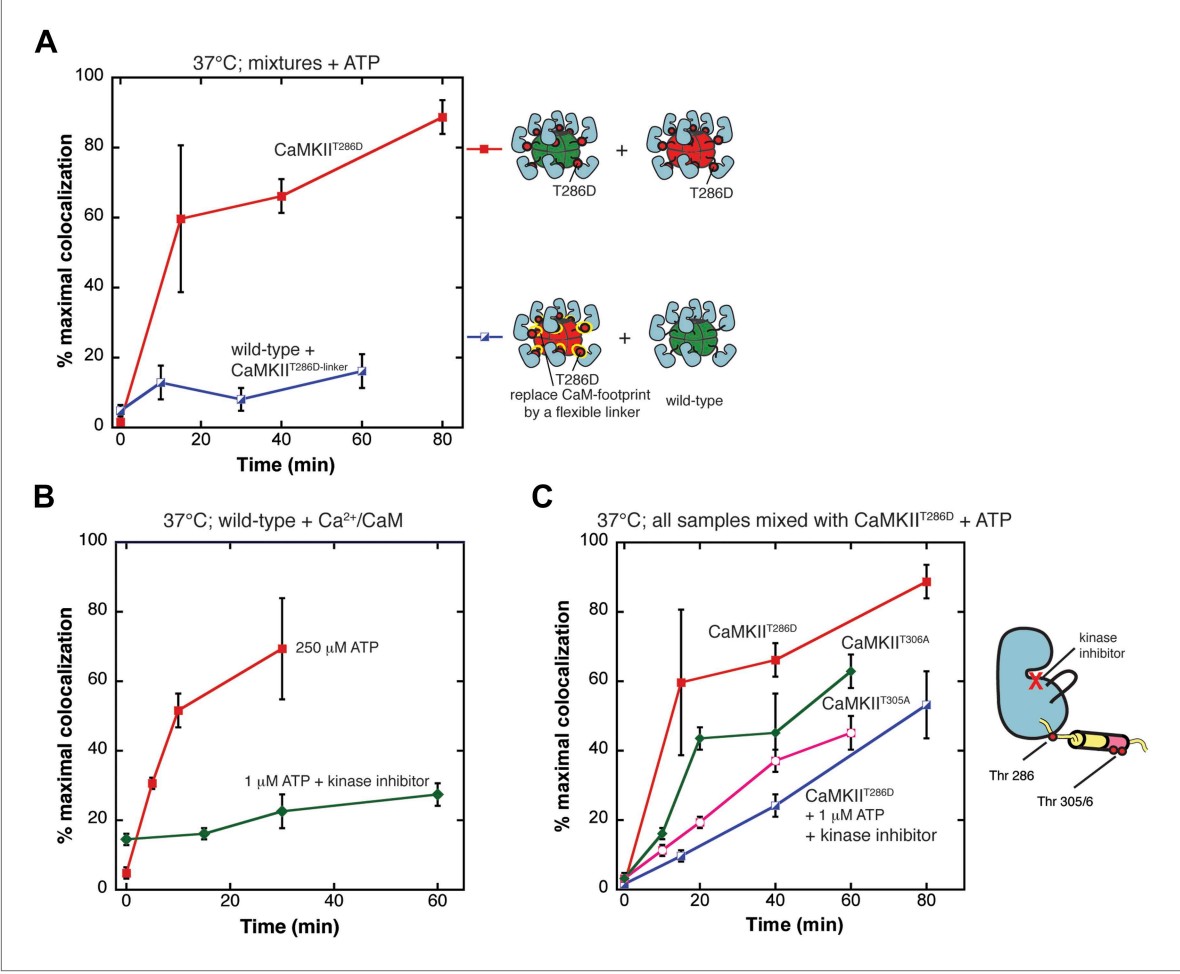

**Figure 4.** Phosphorylation of the calmodulin-recognition element is crucial for exchange. (**A**) Mixing CaMKII[T286D] in the absence of Ca²⁺/CaM results in robust colocalization (red). Mutating the CaM-recognition element (R3) reduces colocalization significantly (blue). (**B**) Both mixing experiments shown use wild-type CaMKII in the presence of Ca²⁺/CaM. The addition of a kinase inhibitor and 1 μM ATP significantly reduces colocalization (green) compared to a condition with full kinase activity (250 μM ATP, no kinase inhibitor) (red). (**C**) All species are mixed with CaMKII[T286D] in the absence of Ca²⁺/CaM. Compared to the colocalization resulting from mixing CaMKII[T286D] (red), mixing CaMKII[T286D] in the presence of a kinase inhibitor results in reduced colocalization (blue). Replacement of either Thr 305 or Thr 306 by alanine also results in a reduction in colocalization (pink and green, respectively).

The following figure supplements are available for figure 4:

**Figure supplement 1**. Kinase activity is crucial for exchange.

the R3 element from the kinase domain, but it remains hidden within calmodulin. Thr 286 is, however, outside the footprint of calmodulin, and it is phosphorylated upon Ca²⁺/CaM binding to the holoenzyme. The release of Ca²⁺/CaM from a subunit after Thr 286 is phosphorylated facilitates phosphorylation at Thr 305 and Thr 306, which prevents Ca²⁺/CaM rebinding (**Hashimoto et al., 1987**; **Hanson and Schulman, 1992**; **Colbran, 1993**). Displacement of Ca²⁺/CaM and phosphorylation of the R3 element on Thr 305 and/or Thr 306 may therefore facilitate subunit exchange. Using this reasoning, the kinase activity of CaMKII is expected to be crucial for the exchange process.

We carried out two mixing experiments using wild-type CaMKII and Ca²⁺/CaM in parallel (**Figure 4B**). In one experiment, as described previously, the fluorescently labeled samples were incubated separately with ATP and Ca²⁺/CaM, then mixed together and monitored for CaMKII colocalization using the single-molecule assay. In the other experiment, the ATP concentration was lowered to 1 μM, which is below the value of the $K_M$ for ATP (**Colbran, 1993**). In addition, the kinase inhibitor bosutinib was added during the incubation phase. Although bosutinib was developed as an inhibitor of Abl tyrosine kinase, it has off-target activity against CaMKII (**Remsing Rix, 2009**) and was used in the

crystallization of autoinhibited CaMKII (*Chao et al., 2011*). The fluorescently labeled samples were then mixed, and monitored for colocalization, as before.

As shown in *Figure 4B*, the reduction in ATP concentration and the presence of bosutinib result in suppression of colocalization. For example, after 30 min of mixing, the wild-type enzyme exhibits 70% of the maximal degree of colocalization, whereas under conditions where phosphorylation is inhibited the level of colocalization is only ~20% after 30 min. We confirmed the importance of kinase activity by using the FRET assay (*Figure 4—figure supplement 1*). In this case, we added $Ca^{2+}$/CaM and the kinase inhibitor in the absence of ATP, and found that there is no FRET signal under these conditions.

We compared the colocalization of the constitutively active variant CaMKII[T286D] in the presence and absence of the kinase inhibitor and low ATP. As shown in *Figure 4C*, inhibition of kinase activity suppresses colocalization for CaMKII[T286D], although to a lesser extent than for the wild-type protein. These data indicate that phosphorylation at sites other than Thr 286 is also important for exchange, and drew attention to a possible role for Thr 305 and Thr 306 in the exchange process. We replaced these residues with alanine separately to generate two constructs (CaMKII[T286D,T305A] and CaMKII[T286D,T306A]). Attempts to purify a construct in which both residues were mutated to alanine were not successful because this variant did not express well. Mixing experiments demonstrated substantial reduction in colocalization for both individual mutations, although somewhat less than the reduction obtained with treatment of CaMKII[T286D] with bosutinib (*Figure 4C*). We conclude that phosphorylation of both Thr 305 and Thr 306, in addition to Thr 286, is important for subunit exchange.

## Activated CaMKII can exchange subunits with holoenzyme assemblies that have not been activated, leading to a spread of activation

An important question is whether subunits that are activated can exchange with holoenzymes that have not been activated, and further, whether such exchange could result in the phosphorylation of Thr 286 in the unactivated subunits. To study this, we used mixtures of wild-type CaMKII and the constitutively active CaMKII[T286D] variant in mixing experiments. Because CaMKII[T286D] lacks Thr 286, this allows us to detect the phosphorylation of Thr 286 in the unactivated holoenzymes by using an antibody that is specific for the phosphorylated form of Thr 286 (pT286, see 'Materials and methods'). The antibody is labeled with Alexa 647, and it does not detect CaMKII[T286D] (*Figure 5—figure supplement 1*).

The first important result is that subunits in constitutively active CaMKII[T286D] can exchange with subunits from holoenzymes that have not been activated, when ATP is added (*Figure 5A*). Fusion of the hexameric protein Hcp1 to the wild-type protein reduces the rate of colocalization, indicating that the unactivated CaMKII holoenzyme has to open up in some way for the exchange process to happen.

We also examined the ability of the activated subunits to phosphorylate subunits from holoenzymes that had not been activated. To do this we devised a three-color single-molecule experiment, in which the Alexa 488 and Alexa 594 channels monitor the locations of the labeled CaMKII variants, and the Alexa 647 channel detects phosphorylated Thr 286 in wild-type CaMKII. To determine the extent of antibody labeling that can be detected in a fully activated sample of CaMKII, we first activated CaMKII by $Ca^{2+}$/CaM and ATP for 10 min. This is expected to result in robust phosphorylation of each holoenzyme assembly because of the direct activation of each subunit by $Ca^{2+}$/CaM. This resulted in ~75% colocalization of fluorescent signals from CaMKII and the pT286 antibody (*Figure 5—figure supplement 1*).

We then studied the extent of Thr 286 phosphorylation that is detected when the constitutively active CaMKII[T286D] is mixed with unactivated wild-type CaMKII. As shown in *Figure 5B*, there is an increase in the degree of phosphorylation detected with time, demonstrating that CaMKII[T286D] is able to phosphorylate subunits in unactivated wild-type CaMKII.

It has been shown previously that trans-phosphorylation of Thr 286 occurs principally within a holoenzyme (*Hanson et al., 1994*; *Rich and Schulman, 1998*). In order to check that the observed phosphorylation is due to the simultaneous presence of activated (CaMKII[T286D]) and unactivated (wild type) subunits within the same holoenzyme, we carried out two analyses. First, we employed the Hcp1 fusion construct, for which subunit exchange was suppressed (*Figure 3A*). Mixing CaMKII[T286D]-Hcp1 with wild-type CaMKII resulted in decreased Thr 286 phosphorylation (*Figure 5B*), indicating that stabilizing the activated CaMKII holoenzyme reduces the observed phosphorylation. In the second analysis, we showed that antibody binding occurs principally in those species in which the red and green fluorophores are colocalized (*Figure 5C*), consistent with subunit exchange being required for the trans-phosphorylation reaction.

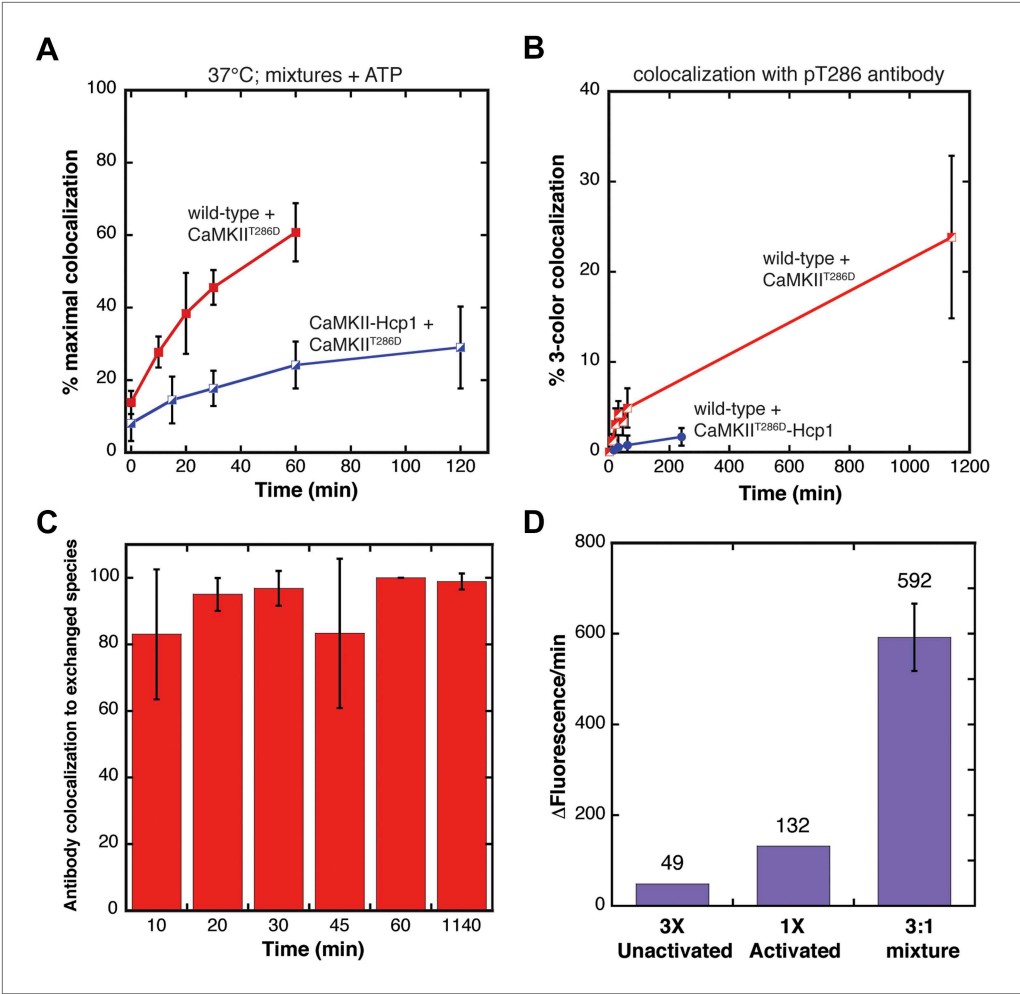

**Figure 5**. Evidence for the spread of phosphorylation into unactivated holoenzymes. (**A**) Both mixing experiments shown use wild-type CaMKII in the presence of $Ca^{2+}$/CaM and ATP. CaMKII$^{T286D}$ mixed with unactivated wild-type CaMKII results in high colocalization (red). This colocalization is suppressed by the addition of the Hcp1 module (CaMKII-Hcp1; blue). (**B**) Levels of pThr286 labeling in each mixing experiment from (**A**). The pThr286 antibody is modified with Alexa 647, which is then added to the mixed samples. Subsequent analysis is for 3-color colocalization between Alexa dyes 488, 594, and 647. The phosphorylation spreads significantly more in the CaMKII$^{T286D}$ sample (red) compared to the sample mixed with CaMKII$^{T286D}$-Hcp1 (blue). (**C**) There is colocalization of the pThr286 antibody with particles that have both Alexa 488 and 594, indicating that the antibody is only binding to those CaMKII holoenzymes that have already exchanged subunits. The graph shows the fraction of antibody label that is colocalized to particles that contain both the red and green fluorophores. Note that this fraction is close to 100%. (**D**) Kinase activity against a peptide substrate (syntide) was monitored in solution using the ADP Quest assay, where the fluorescence of resorufin is an indicator of ATP consumption. The reaction rates are plotted for three separate samples. First, 3 µl of an unactivated CaMKII sample, then 1 µl of an activated CaMKII sample (both are at the same final protein concentration), and finally a mixture of these components. The value of the reaction rate is indicated above each bar. It is clear that the reaction rate in the mixture is higher than just the addition of the rates of the individual components.

The following figure supplements are available for figure 5:

**Figure supplement 1**. Controls for the pThr286 antibody.

We note that the signal from pT286 rises much more slowly than the rate of colocalization. Colocalization reaches a plateau value of ~60% colocalization in ~50 min, whereas the level of three-color colocalization at a comparable time point is only ~5%. The replacement of Thr 286 by aspartate in CaMKII$^{T286D}$ may not be a good mimic of the phosphorylated form of Thr 286 in terms of its ability

to promote phosphorylation. We found that the activity of CaMKII[T286D] towards a peptide substrate (syntide, see 'Materials and methods') is only ~20% that of activated wild-type CaMKII (data not shown). Also, the ability of CaMKII[T286D] to transphosphorylate an adjacent subunit within an unactivated holoenzyme may be inherently slow without calmodulin bound to the subunit being phosphorylated (*Rich and Schulman, 1998*). The activation of CaMKII is highly cooperative (*Gaertner et al., 2004*; *Rosenberg et al., 2005*), so low levels of Ca$^{2+}$/CaM that are insufficient for autophosphorylation of an unactivated holoenzyme may be sufficient for binding the target subunit and enhancing its auto-phosphorylation in holoenzymes that contain some activated subunits through exchange.

To get a better sense of the extent to which activated subunits can potentiate the activity of holoenzymes that have not been activated, we turned to a solution assay using two different samples of wild-type CaMKII, without fluorescent labeling. One sample, the activated sample, was prepared by incubation with ATP and Ca$^{2+}$/CaM for 15 min, followed by removal of Ca$^{2+}$/CaM (see 'Materials and methods'). The second sample, the unactivated sample, was not exposed to Ca$^{2+}$/CaM. We then measured the activity of these samples using a peptide substrate (syntide) and a continuous enzyme-coupled assay (ADP Quest, see 'Materials and methods'). This assay relies on the conversion of ADP to a fluorescent product, resorufin, the accumulation of which is monitored over time (*Charter et al., 2006*).

A sample containing 1 µl of activated CaMKII at 6 µM concentration exhibits a peptide phos-phorylation rate of 132 units, where the units are the change in the fluorescence of resorufin in unit time (*Figure 5D*). A second sample containing 3 µl of unactivated CaMKII at the same concentration exhibits a rate of 49 units (this activity represents the futile ATP hydrolysis rate of CaMKII, also observed without peptide substrate, data not shown). A third sample consisting of 1 µl of activated CaMKII at 6 µM mixed with 3 µl of unactivated CaMKII at 6 µM exhibits a rate of 592 units. In the absence of any crosstalk between the unactivated and activated holoenzymes, the reaction rate exhibited by the mixture is expected to be the sum of the reaction rates of the first two samples, that is, 181 units. The actual observed rate, 592 units, is about three times as high. This suggests that activated CaMKII holoenzymes have increased the activity of the holoenzymes that had not been activated previously.

## Speculation about mechanisms for subunit exchange

One difficulty in developing a conceptual model for the exchange process in CaMKII is that there appears to be little precedent in the biochemical literature, as far as we are aware, for oligomeric enzymes that spread their state of activation by exchanging subunits. We envisage two possible kinds of mechanisms for the exchange of subunits between CaMKII holoenzymes. The first mechanism is suggested by the unusual allosteric activation of porphobilinogen synthase (*Selwood and Jaffe, 2012*). This enzyme converts between an inactive hexamer and an active octamer, with the transition requiring dissociation of the assembly into dimers (*Selwood et al., 2008*). We have developed a spec-ulative structural model for how activation might promote the release of subunits, most likely dimers, from a CaMKII holoenzyme, discussed below. An alternative mechanism involves the transient formation of higher order aggregates of two or more holoenzymes, with exchange occurring within the aggregate followed by separation of holoenzymes with mixed subunits. The data available to us at present provide no clues as to how such a process might occur, so we do not discuss this further although it remains an important alternative mechanism for future investigation.

Regardless of whether subunit exchange proceeds through the release of subunits or through transient aggregation, it seems likely that the regulatory segment is able to destabilize the hub assembly when this segment is released from the kinase domain. We developed a model for how the regulatory segment might weaken the hub assembly by analyzing molecular dynamics simulations of CaMKII (see 'Appendix'), and by considering several characteristic features of the structure of the holoenzyme and the exchange process. These include (i) the ability of the hub assembly to interconvert between dodecameric and tetradecameric forms, (ii) the importance of phosphorylation of the regu-latory segment, (iii) the presence within the hub domain of a deep cavity containing three conserved arginine residues and (iv) the known ability of the hub domain to serve as a docking site for peptide segments by virtue of its exposing an uncapped β sheet, with no steric hindrance to extending the sheet by an additional strand provided by the regulatory segment. For ease of discussion, the details of the molecular dynamics simulations and the reasoning that went into the development of this model are presented in an 'Appendix' that follows immediately after the main body of the text. The main features of the model are summarized schematically in *Figure 6*. Note that the schematic in *Figure 6*

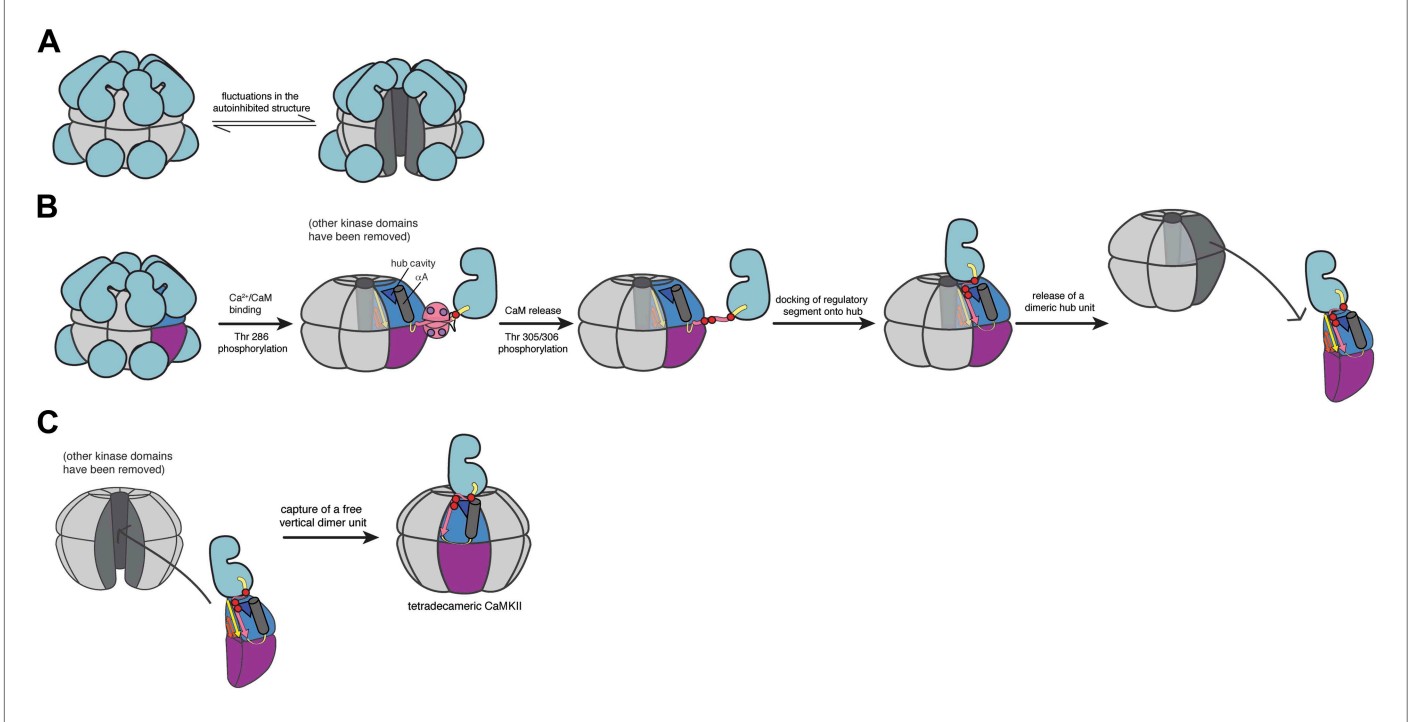

**Figure 6**. Schematic of a potential mechanism for subunit exchange in CaMKII. (**A**) In the unactivated state of CaMKII, the dodecameric hub domain undergoes fluctuations, which leads to transient lateral openings between vertical dimeric units. (**B**) Upon Ca²⁺/CaM binding, Thr 286 is trans-phosphorylated. For simplicity, just one kinase domain is depicted. When calcium levels drop, CaM falls off and Thr 305 and Thr 306 are subsequently phosphorylated. The now-released regulatory segment is free to bind the open β sheet of its own hub domain. This brings residues 305/306 in close proximity to the hub cavity (blue triangle), which houses the three conserved Arg residues. Binding of the regulatory segment induces a crack to open in the hub domain, which exposes the Arg residues to the phosphate groups. We reason that phosphorylation of the regulatory segment leads to an interaction between the R3 element and the hub domain that weakens the lateral association between hub domain dimers, leading to their release from one holoenzyme. (**C**) Fluctuations in the autoinhibited holoenzyme create a hub assembly that resembles a 'C' shaped structure. This fluctuation may allow the capture of a vertical dimer that has been released from an active holoenzyme. The drawing depicts the adoption of a tetradecameric structure upon docking of an incoming vertical dimer, which may be an intermediate in the exchange process. As discussed in the main text, we have no direct experimental evidence at present concerning the exchange process. We cannot, therefore, rule out alternative mechanisms, such as those involving a transient aggregation of holoenzymes prior to exchange.

emphasizes the release of dimers, but we are uncertain as to whether the actual process involves a transient aggregation step that we do not yet understand.

The CaMKII holoenzyme can be thought of as being assembled from six 'vertical dimers'. In a view that has the mid-plane of the dodecamer horizontal, these vertical dimers each contribute one hub domain to the upper ring and one to the lower ring of the hub assembly. We propose that the hub assembly within a holoenzyme, whether activated or not, normally undergoes fluctuations that convert it from the closed double-ring form seen in crystal structures to an open C-shaped form with a gap between two adjacent vertical dimers. Molecular dynamics simulations suggest that this gap can easily be large enough to capture a vertical dimer provided by another holoenzyme. This could, for example, convert a dodecameric assembly to a tetradecameric one. In the absence of activation, however, the transient openings in the hub assembly of the holoenzyme simply reanneal, due to the stability of the lateral interactions between vertical dimers.

The situation changes, however, when CaMKII is activated and Thr 305 and Thr 306 are phosphorylated. We propose that the phosphorylated regulatory segment, which is freed from interaction with the kinase domain and Ca²⁺/CaM, is now able to dock on the hub domain and form an additional strand of the β sheet that is the core scaffold of the hub domain. We speculate that when this happens, one or more of the phosphorylated sidechains in the regulatory segment reach into the interior of the hub and interact with the conserved arginine residues located in the hub cavity. This interaction might distort the structure of the hub domain, slightly weakening the lateral interactions in the hub assembly.

A fluctuation in the hub assembly can now result in the release of an activated vertical dimer. A released vertical dimer can either be recaptured or, if it encounters an unactivated holoenzyme that is open transiently, it can be incorporated into that holoenzyme.

While we believe that this model provides a reasonable framework for thinking about the exchange process, the schematic shown in *Figure 6* is far from definitive. For example, close encounters between two holoenzymes may be necessary for an activated dimer to 'hop' directly from one holoenzyme to another, without being ever released completely. In such a situation, the interaction between the phosphorylated regulatory segment and the hub domain may occur in *trans*, between two holoenzymes, rather than within one holoenzyme. We anticipate that future studies will establish whether the model we are proposing is correct in essence, or whether the subunit exchange mechanism relies on features that we have failed to anticipate.

## Concluding remarks

The strengthening of synaptic connections between neurons that occurs following brief repetitive stimuli that induce LTP is likely to be important for the formation and maintenance of cognitive memory. Much attention has been focused on the role of CaMKII in this process, particularly its ability to maintain an activated state, through intra-holoenzyme phosphorylation of Thr 286, after cessation of LTP-inducing stimuli. But can the active state be maintained beyond the lifetime of the kinase molecules present during the initial stimulus?

Lisman introduced the idea that information about previous activating events could be transmitted to newly synthesized CaMKII molecules if the subunits of the activated holoenzyme could exchange with unactivated ones (*Lisman, 1994*). We now demonstrate that the CaMKII holoenzyme has precisely this property when studied in vitro. Activated CaMKII holoenzymes can indeed instruct previously unactivated holoenzymes as to their phosphorylation state through subunit exchange, and this ability is switched on by activation and autophosphorylation. In the time since Lisman's earlier speculation about subunit exchange, there has been no experimental precedent that has pointed to such a mechanism being operative for CaMKII. Translocation of CaMKII to the NMDA receptor requires active subunits and has been postulated to serve as a molecular memory (*Bayer et al., 2001*; *Jiao et al., 2011*; *Lemieux et al., 2012*; *Neant-Fery et al., 2012*). This population of CaMKII bound to the NMDA receptor may maintain its activity for longer times (*Otmakhov et al., 2004*). Subunit exchange could facilitate the recruitment of additional holoenzymes to the receptor by spreading activation to unactivated holoenzymes.

Recent experiments using a CaMKII construct engineered to be a FRET reporter of its activation state indicate that the bulk of CaMKII holoenzymes in dendrites maintain their activation state for not much longer than a minute after the stimulation is withdrawn (*Takao et al., 2005*). Interestingly, another study demonstrated that localized stimulation of a neuron resulted in the translocation of CaMKII from the main body of the axon to synapses throughout the dendritic arbor for much longer times (15–40 min) after the local stimulation is stopped; the authors of this report commented that subunit exchange may play a role in this process (*Rose et al., 2009*). Given the high concentrations of CaMKII in synapses and the fact that calmodulin is limiting (*Liu and Storm, 1990*; *MacNicol and Schulman, 1992*; *Persechini and Stemmer, 2002*), subunit exchange provides a potential mechanism for increasing the spread of activation caused by an initial activating pulse.

Now that we have demonstrated unambiguously that activation triggers the exchange of subunits in CaMKII in vitro, future experiments will address several important questions. We have focused on human CaMKIIα in this study. Is subunit exchange common to the other isoforms of the human enzyme? Did the property of subunit exchange arise early in evolution, or did it evolve with the specialization of CaMKII to neuronal function? Finally, and most importantly, how has nature exploited subunit exchange in activated CaMKII to determine the actual outcomes of neuronal signaling as well as in heart and other systems? It will be challenging, but ultimately most informative, to devise experiments to address this last question.

## Appendix: speculation concerning the activation-dependent weakening of the hub assembly of a CaMKII holoenzyme

The CaMKII dodecamer can be described in terms of the lateral association of six vertical dimeric units, as shown in *Figure 7*. We imagine that a critical step in the exchange process is a lateral opening of the ring formed by the hub domains (*Figure 7A,B*). The exchange process might involve the release

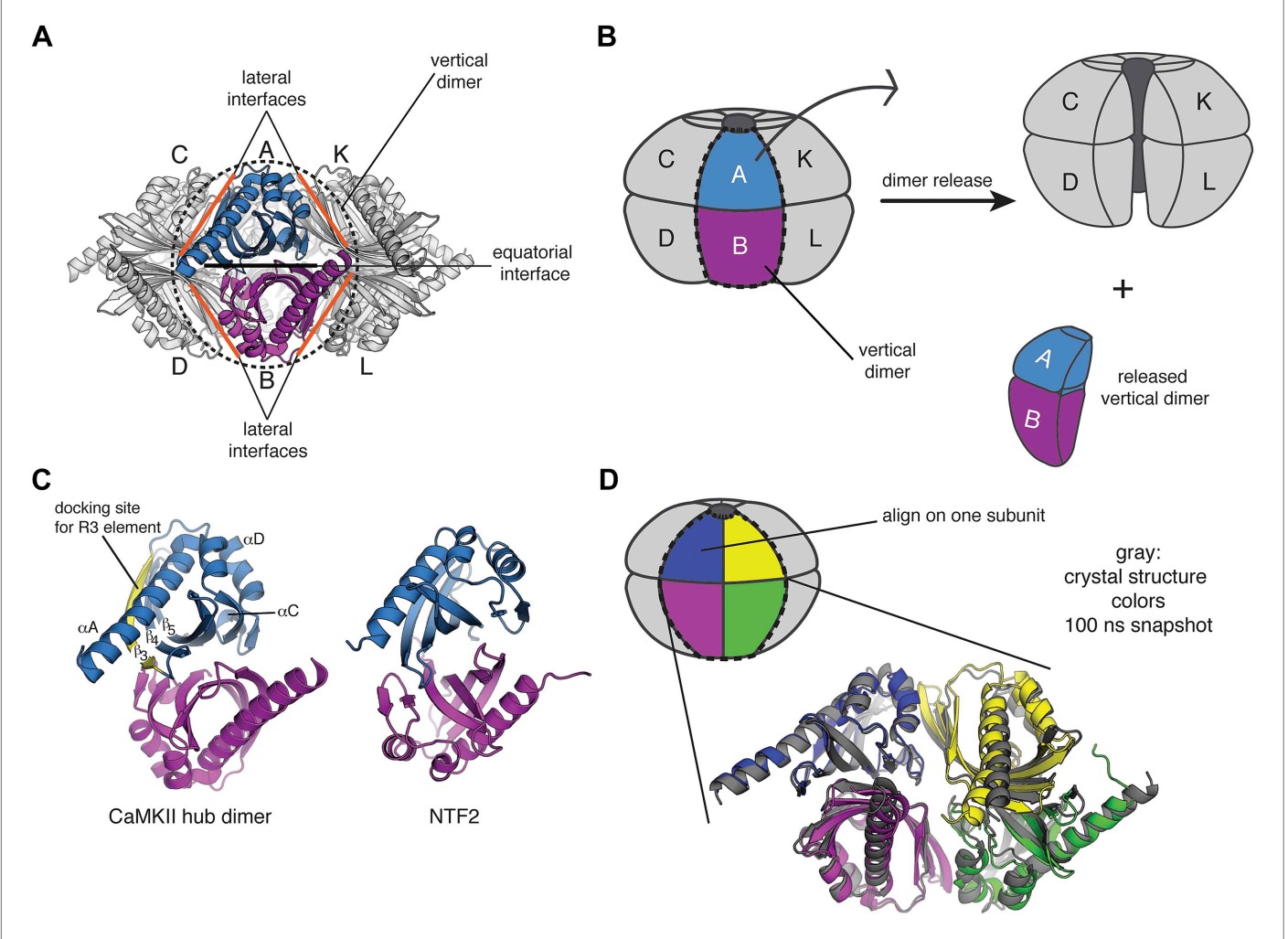

**Figure 7**. A vertical dimeric unit of the CaMKII assembly may be the unit of exchange. (**A**) The CaMKII hub assembly can be described as a set of six vertical dimers, and each dimer is labeled as A/B; C/D; etc. One of these dimers (A/B) is highlighted in blue/magenta (black dashed line). The lateral interfaces and equatorial interface for this dimer are indicated by orange and black lines, respectively. (**B**) A schematic diagram that indicates how one vertical dimer may be released from the holoenzyme. (**C**) The structure of the vertical hub dimer from CaMKII is shown in comparison to a dimer of NTF2 (PDB codes: 2UX0 and 1OUN, respectively). The notation for the secondary structural elements of the CaMKII hub domain are shown. (**D**) A molecular dynamics simulation was started from the dodecameric crystal structure of the hub domain (PDB code: 2UX0). The starting crystal structure is overlaid with an instantaneous structure from the molecular dynamics trajectory at 100 ns, aligning onto just one subunit (indicated on the schematic). It is clear that the vertical dimeric unit is relatively stable (blue/magenta), but there is a significant change in the relative positioning of the blue/magenta dimer with respect to the yellow/green dimer. This indicates that the lateral interfaces are more dynamic than the equatorial interfaces.

The following figure supplements are available for figure 7:

**Figure supplement 1**. Molecular dynamics simulations suggest that the contacts across equatorial interfaces are stronger than those across the lateral interfaces.

of vertical dimers. Such a vertical dimer could not be released easily from the Hcp1 fusion construct and indeed subunit exchange is suppressed between these variants (***Figure 3A***). The release and capture of vertical dimers also provides a route for the interconversion of dodecameric and tetradecameric holoenzymes without requiring a more complete disassembly. As noted in the main text, we are uncertain about whether the transient aggregation of holoenzymes might be necessary for exchange to occur. In this Appendix, we focus on potential interactions between the regulatory segment and a single hub domain assembly.

That the unit of exchange might be a vertical dimer is suggested by the fact that the hub domain is very closely related in structure, although not in sequence, to members of a large family of enzymes and binding proteins, such as nuclear transport factor 2 (NTF2) and ketosteroid isomerase like proteins, which we refer to as the NTF2 family (*Figure 7C*; *Hoelz et al., 2003*). A common quaternary structural unit within the NTF2 family corresponds to the vertical dimer in the CaMKII hub. Of the top 50 unique hits in a Dali search (*Holm and Rosenstrom, 2010*) for structures similar to the mouse CaMKIIα hub domain (PDB code: 1HKX), 29 form dimers in the crystal lattice that are similar to the vertical CaMKII hub dimer (*Figure 7C*).

## Molecular dynamics simulations suggest that the dodecameric hub assembly is strained

Molecular dynamics simulations provide a clear indication that the lateral interfaces in the hub assembly are more likely to be disrupted than the equatorial ones (*Figure 7—figure supplement 1*). The hub domain of human CaMKIIγ has been crystallized as a dodecameric assembly (*Rellos et al., 2010*) (PDB code: 2UX0), and we generated a 100 nanosecond (ns) trajectory of this assembly. The internal motions of the hub can be described as rigid-body motions of individual vertical-dimer units, with alterations in the lateral dimer–dimer interfaces. This is demonstrated by aligning each instantaneous structure from the trajectory on each subunit in turn, and graphing the root mean square deviations in the positions of the Cα atoms in the two subunits across the lateral interfaces and the one subunit across the equatorial interface (*Figure 7D*, *Figure 7—figure supplement 1*). For comparisons across the equatorial interfaces, within a vertical dimer, the rms displacements in the neighboring subunits are ~2 Å across the ring. In contrast, for comparison across the lateral interfaces, the rms displacements of the neighboring subunits are as much as 6 to 10 Å for several of the interfaces, consistent with the lateral interfaces being much more flexible than the equatorial ones (*Figure 7—figure supplement 1*).

Further analysis of the simulation suggests that there is strain associated with the closed ring formed by the dodecameric hub assembly of CaMKII. Although the hub assembly is symmetric initially, the sixfold symmetry of the ring breaks down during the course of the simulation. This is due to the tightening of some of the lateral interfaces, for which the buried surface area increases. For the CaMKIIγ structure used in the simulation, the closer packing involves the sidechains of Phe 364, Phe 367, Tyr 368, Asn 371, Leu 372 on helix αD in one subunit and Gln 406, Pro 414, Thr 416 and Ile 418 (Gln 418 in the α isoform) from strands β4 and β5, as well as Pro 379 in strand β3 in the other subunit (see *Figure 7C* for the notation). With the exception of Ile 418, these residues are all conserved in CaMKIIα.

The tighter packing at some of the interfaces in the simulation of the dodecameric hub assembly is coupled to looser packing at some of the other interfaces. At one of the interfaces, the residues on helix αD in one subunit and the β4 and β5 strands on the other are splayed apart so that the sidechains of Phe 364 and Phe 367, which are buried in the more stable interfaces, are now partially solvent exposed (*Figure 8*). These results suggest that the constraint of ring closure prevents the simultaneous optimization of all of the lateral interfaces.

To further explore the extent to which the closed-ring form of the dodecamer is strained, we removed one vertical dimer from the crystal structure of the dodecamer, generating a C-shaped decameric model (*Figure 9A*). This allows us to use molecular dynamics to study the effect of relaxing the ring-closure constraint, following a strategy that was used effectively to analyze strain in circular sliding DNA clamps (*Jeruzalmi et al., 2001*; *Kazmirski et al., 2005*).

This decameric structure, derived from the crystal structure of the dodecameric CaMKIIγ hub assembly, was used to initiate two independent molecular dynamics trajectories (100 ns and 50 ns, respectively). There is a rapid relaxation of the curvature of the ring in both trajectories. The decameric structure opens up with respect to the starting structure in both trajectories within ~10 ns (*Figure 9B*). The principal outward displacement occurs in the plane of the ring so that the structures move closer to the tetradecameric form of the hub assembly (PDB code: 1HKX; *Figure 9B*, *Figure 9—figure supplement 1*).

The constraint of ring closure undoubtedly imposes an entropic strain on the ring, and removal of the constraint is expected to yield a more floppy and open structure. What is notable, however, is that in both molecular dynamics trajectories we observe relaxation to a specific interfacial arrangement that is different from the interfacial arrangement seen in the crystal structure of the dodecamer. There are two internal vertical interfaces in the decamer, between the G:H and E:F vertical dimers

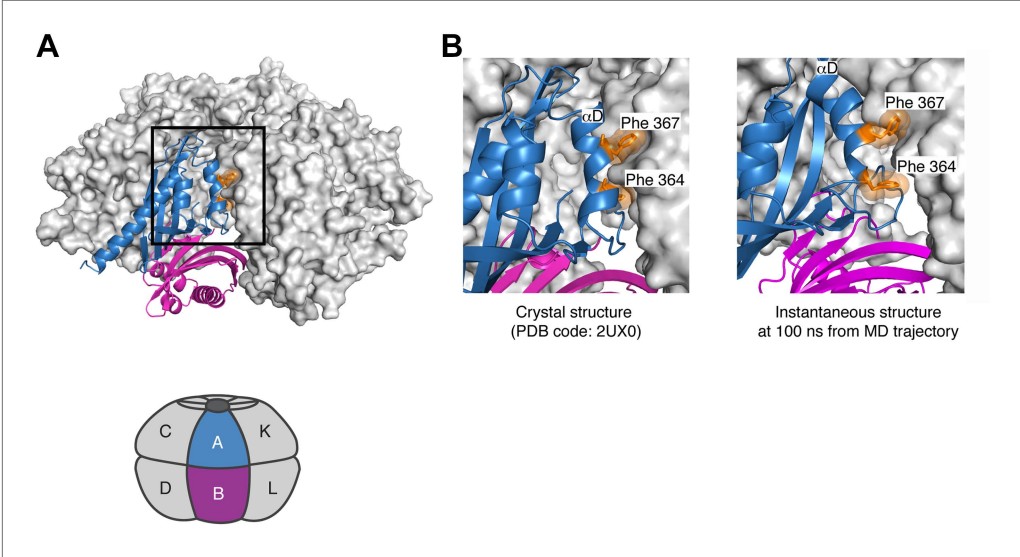

**Figure 8**. Strain associated with the closed ring formed by the dodecameric hub assembly of CaMKII. During the simulation of the CaMKIIγ dodecamer, the sixfold symmetry of the hub assembly breaks down due to the tightening of some of the lateral interfaces and loosening at others. (**A**) A view of one of the lateral interfaces, with a close-up view in (**B**). At this interface, the residues on helix αD in one subunit and the β4 and β5 strands on the other are splayed apart after 100 ns of simulation, so that the sidechains of Phe 364 and Phe 367, which are buried in the crystal structure (left) and more stable interfaces, are now partially solvent exposed (right). These results suggest that the constraint of ring closure prevents the simultaneous optimization of all of the lateral interfaces.

and between the G:H and I:J vertical dimers. The relative orientations of subunits across these two interfaces converge to a similar arrangement, with low rms displacements, both within one trajectory and between two independent trajectories (*Figure 9C*, bottom left, *Figure 9—figure supplement 2A*), and this arrangement is different from that seen in the crystal structure of the dodecamer (*Figure 9C*, bottom right, *Figure 9—figure supplement 2B*).

This relaxation towards a specific structure, as seen in two independent trajectories, suggests that there is an enthalpic contribution to the strain in the closed ring. It is difficult, however, to assign this strain to any particular sets of interaction. The structural relaxation preserves the set of residues that interact at each interface, and PISA analysis (*Krissinel and Henrick, 2007*) of instantaneous structures extracted from the trajectories does not reveal any systematic trend in the estimated interfacial free energy.

## Modeling a possible interaction involving the phosphorylated regulatory segment and the hub assembly

The experimental data suggest that the release of the regulatory segment from the body of the kinase domain upon activation allows the regulatory segment and, possibly the kinase domain, to interact with the hub domain in a way that weakens the inter-subunit interfaces in the hub assembly. Given that the calmodulin-binding R3 element is particularly important for exchange, we looked for plausible ways in which the phosphorylated R3 element could interact with the hub domain.

There is one very obvious binding site in the hub domain for negatively charged groups, and that is the deep cavity in the hub domain, which corresponds to the active site or peptide-binding site in other structurally related proteins (*Hoelz et al., 2003*). We shall refer to this as the 'hub cavity' (*Figure 10A,B*). CaMKIIα contains three arginine residues that are located deep within the hub cavity (residues 403, 423, and 439). These residues are highly conserved either as arginine or lysine, with three positively charged residues present in CaMKII from *C. elegans*, mouse, and drosophila. No function has been ascribed to the hub cavity in CaMKII nor to the arginine residues within it.

A feature of the hub assembly that might allow interaction between Thr 305 and Thr 306 in the regulatory segment and the arginine residues in the interior of the hub cavity is suggested by an interesting variability in the structures of individual subunits of the hub assembly in crystal structures.

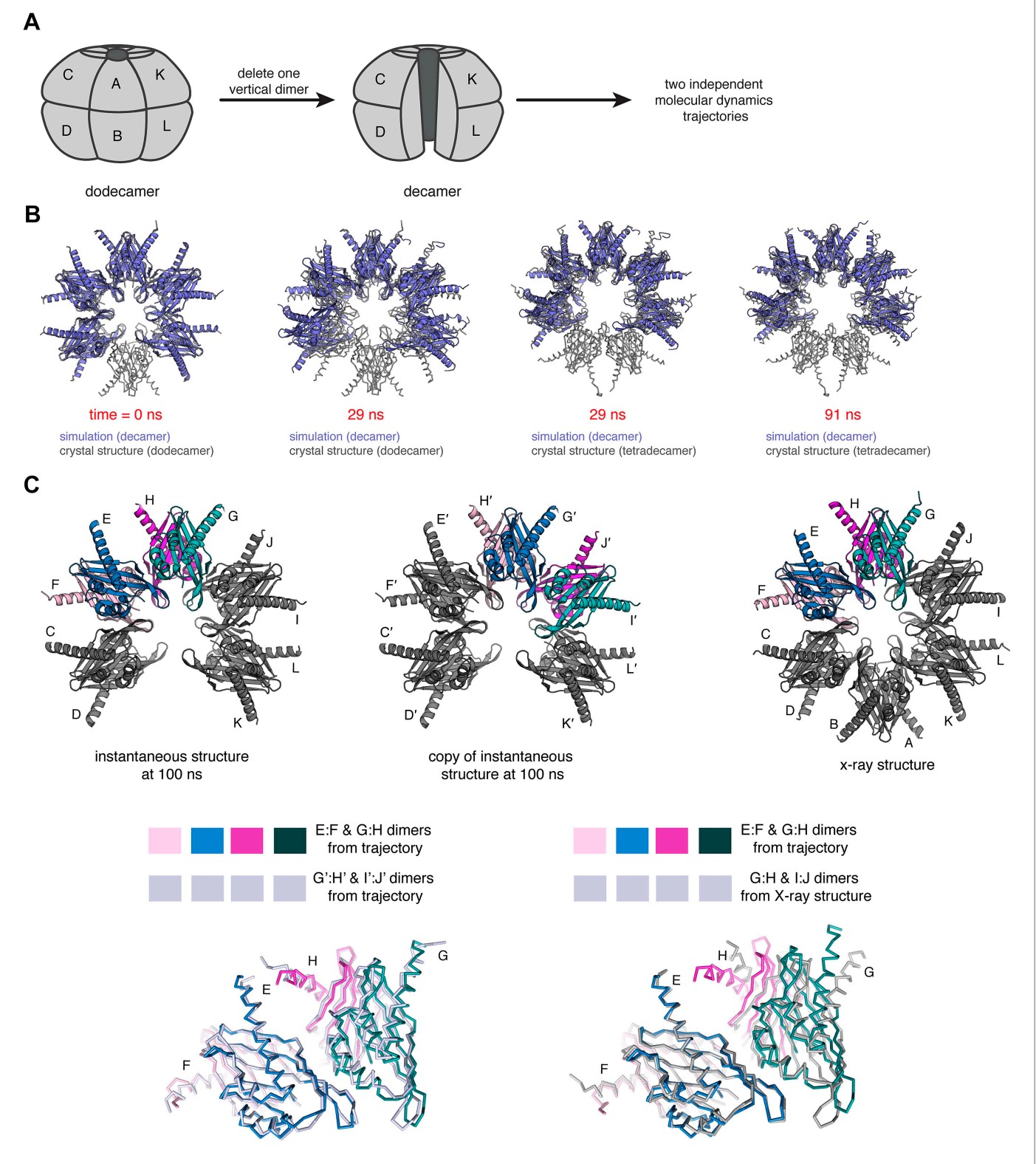

**Figure 9**. Molecular dynamics simulations of an open-ringed (decameric) hub assembly. (**A**) The dodecamer consists of six vertical dimers, denoted A:B, C:D…K:L. The decamer used in the simulations is created by removing the A:B dimer. (**B**) We initiated two independent molecular dynamics trajectories from this decameric structure and the results for one are shown in this diagram. Instantaneous structures from this simulation are shown overlaid with the crystal structure for either the dodecameric (PDB code: 2UX0) or tetradecameric (PDB code: 1HKX) hub assembly. At 29 ns, it is clear that the decamer

*Figure 9. Continued on next page*

*Figure 9. Continued*

has relaxed to the tetradecameric conformation, with further opening evident at 91 ns. (**C**) There are two internal vertical interfaces in the decamer, between the E:F and G:H vertical dimers and between the G:H and I:J vertical dimers (colored in the structural diagram shown at the top). To demonstrate the convergence of the two vertical interfaces to an arrangement that is distinct from the interfaces in the crystal structure, we calculated the displacement of atomic positions in interfacial subunits after the E:F/G:H and G:H/I:J interfaces were brought into spatial alignment using only one subunit, for a series of instantaneous structures extracted from the trajectories. To do this, we made a copy of the trajectory. The subunits in the original are labeled C through K, and in the copy they are labeled C' through K'. The two internal interfaces (E:F/G:H and G:H/I:J) are brought into alignment by superimposing subunit E from the original onto subunit G' of the copy of the trajectory, for pairs of structures at the same point in the trajectory. The overlaid structures are shown at the bottom left. The close overlap shows that the two vertical interfaces in this instantaneous structure are similar. We then aligned subunit E of the instantaneous structure with subunit G of the crystal structure (bottom right). The poor overlap between the crystal structure and the instantaneous structure is evident.

The following figure supplements are available for figure 9:

**Figure supplement 1**. Relieving strain in the dodecameric ring tends towards a tetradecameric conformation.

**Figure supplement 2**. Structural changes during molecular dynamics, showing the relaxation of the decamer to a specific interfacial arrangement.

Although in many cases the only access to the interior of the hub cavity is through the main opening to the cavity, which is far from where the regulatory segment connects to the hub, some individual subunits in hub domain assemblies exhibit cracks in the surface at the interface between helix αA and the β sheet against which it is packed (*Figure 10B*). These cracks make Arg 403, located within the cavity, partially accessible from the side of the hub domain. If such cracks were to open up further then phosphorylated residues in the R3 element might gain access to the interior of the cavity from the side of the hub domain, in the vicinity of the lateral interfaces between vertical hub domain dimers in the holoenzyme.

We carried out a 1 µs molecular dynamics simulation of a vertical hub domain dimer that was extracted from the crystal structure of mouse CaMKIIα (PDB code: 1HKX, *Video 1*). This simulation revealed transient fluctuations in the hub domain structure that opened the interface between helix αA and the underlying β sheet substantially. The simulation suggests that if the R3 element was to dock alongside the lateral interface then it might be positioned to insert phosphorylated sidechains into the interior of the hub domain when such a transient fluctuation occurs.

The molecular details for such a docking mechanism are suggested by the presence of an open edge on the β sheet of the hub domain right alongside the site where the transient openings into the hub cavity occur. This open edge of the β sheet is in fact utilized for a docking interaction by an 8 residue portion of the R3 element in the structure of autoinhibited CaMKII holoenzyme, in which this portion of the R3 element extends the β sheet by one strand (*Chao et al., 2011*). In that structure, however, further interaction between the R3 element and the hub domain is prevented by the fact that the kinase domain sequesters the rest of the regulatory segment.

We modeled the regulatory segment, with Thr 305 phosphorylated, as an additional strand that extends the β sheet of the hub domain, as shown in *Figure 10C* (see 'Materials and methods' for details of the molecular docking). This model was used to generate three independent molecular dynamics trajectories, each extending for 50 ns. In each of these trajectories the R3 element retains the hydrogen bonding pattern that is consistent with its incorporation into the β sheet of the hub domain and the phosphate group maintains its proximity to the sidechain of Arg 403, suggesting that this interaction is plausible (*Figure 10C*).

We analyzed these 'docked' trajectories for fluctuations that would disturb the contacts made within the hub assembly. The peptide stays docked onto the β3 strand of the hub domain throughout the trajectories, which, in turn, prevents the crack between helix αA and the β sheet from closing (*Figure 10D*). Notably, the widening of the crack is coupled to changes in the disposition of the loops that make lateral contacts with the adjacent vertical dimers (*Figure 10D*). This disruption may be sufficient to allow the release of a vertical dimer from the hub assembly (see *Figure 6* for a schematic representation of such a process).

Experimental validation of this model will require an extensive set of studies that are beyond the scope of this paper. For example, we mutated the arginine residues in the hub cavity that we have implicated in the exchange process, but this resulted in a substantial loss of fluorophore labeling at

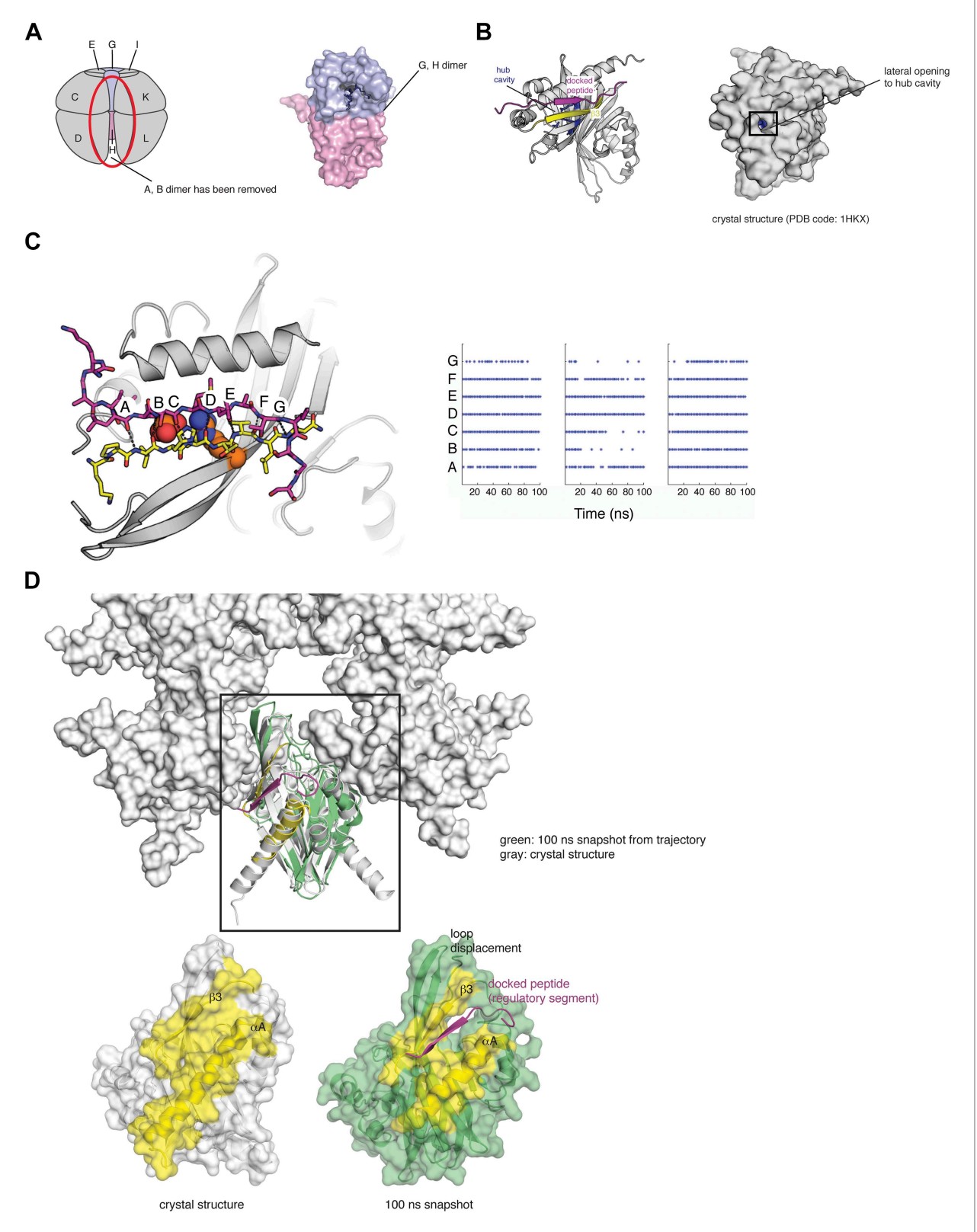

**Figure 10**. Side entrance to the hub cavity and docking of the regulatory segment onto the hub. (**A**) One vertical dimer unit has been removed from the side view of the decameric hub assembly (shown as a schematic on the left). There is limited access to this cavity in the context of the holoenzyme. A close up of the G/H dimer from the crystal structure of CaMKIIγ (PDB code: 2UX0) is highlighted in blue/pink (right). There are three arginine residues

*Figure 10. Continued on next page*

*Figure 10. Continued*

that are located deep within the hub cavity (residues 403, 423, and 439), and these are highly conserved in CaMKII. (**B**) Shown is a vertical dimer from the mouse CaMKIIα crystal structure (PDB code: 1HKX). The crystal structure is shown in cartoon representation (left) with the regulatory segment (magenta) docked onto β3 (yellow). On the right, the crystal structure is shown as a surface representation where the hub cavity that contains the arginine residues (blue) is apparent through a small crack (between αA and β3), which we refer to as the lateral opening. (**C**) The regulatory segment (magenta) was docked onto β3 in the hub assembly (yellow). This docked model was used to generate three independent molecular dynamics trajectories (100 ns each). In each of these trajectories the R3 element retains the hydrogen bonding pattern (right) that is consistent with its incorporation into the β sheet of the hub domain and the phosphate group maintains its proximity to the sidechain of Arg 403, suggesting that this interaction is plausible. Each trajectory was sampled every 2 ns and each dot represents the formation of a hydrogen bond (<3.5 Å). (**D**) Peptide binding disrupts lateral contacts within the hub domain. Docking of the regulatory segment peptide (magenta) prevents the crack between αA (yellow) and β3 (yellow) from closing. This is coupled to changes in the loops that make lateral contacts with the adjacent vertical dimers. This disruption may be sufficient to allow the release of a vertical dimer unit from the hub assembly and facilitate the exchange of subunits between holoenzymes.

residue 335, indicating that the structure of the hub domain might be compromised. A systematic test of the predictions of this model will be carried out in future studies.

## Materials and methods

### Molecular biology

Full-length constructs of human CaMKII were cloned in a pSMT-3 vector containing an N-terminal sumo expression tag (LifeSensors, Malvern, PA). Mutants were generated using a Quikchange protocol (Agilent Technologies, Santa Clara, CA). The linker connecting Hcp1 to CaMKII (GGC GCG TCT GGC GCG TCT GGC GCG TCT) and the hub domain construct were made using standard PCR techniques.

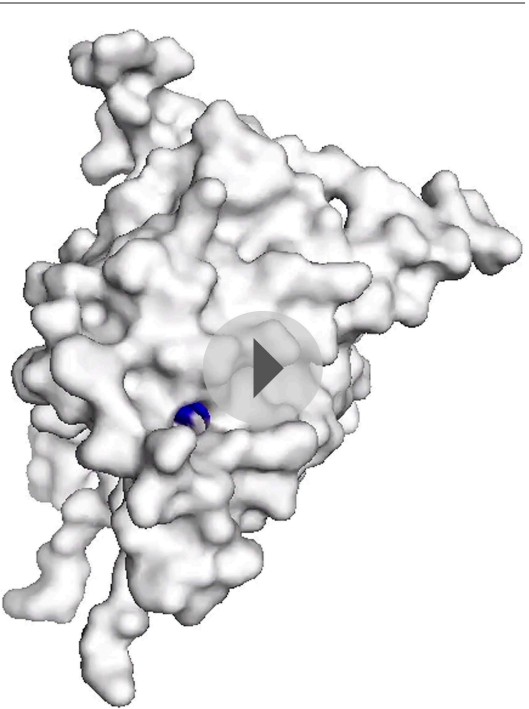

**Video 1**. Lateral opening in the hub domain. Video of a 1 µs molecular dynamics simulation of a vertical hub domain dimer that was extracted from the crystal structure of mouse CaMKIIα (PDB code: 1HKX). This simulation revealed transient fluctuations in the hub domain structure that opened the interface between helix αA and the underlying β sheet substantially. These fluctuations provide access to the hub cavity.

### Protein expression and purification

All CaMKII variants were expressed using *E. coli* and prepared as described previously (*Chao et al., 2010*). Briefly, protein expression was done in Tuner (DE-3)pLysS cells that contained an additional plasmid for λ phosphatase production. Cells were induced by addition of 1 mM isopropyl **β**-D-1-thiogalactopyranoside and grown overnight at 18°C. Cell pellets were resuspended in Buffer A (25 mM Tris, pH 8.5, 150 mM potassium chloride (KCl), 1 mM DTT, 50 mM imidazole, and 10% glycerol) and lysed using a cell disrupter. All purification steps were carried out at 4°C and all columns were purchased from GE Healthcare (Piscataway, NJ). Cleared lysate was loaded on 5 mL Ni-NTA column, eluted with 0.5 M imidazole, desalted using a HiPrep 26/10 desalting column into Buffer A with 0 mM imidazole, and cleaved with Ulp1 protease (overnight at 4°C). The cleaved samples were loaded onto the Ni-NTA column and the flow through was loaded onto a Q-FF 5 ml column, and then eluted with a KCl gradient. Eluted proteins were concentrated and then buffer-exchanged using a Superpose 6 gel-filtration column equilibrated in 25 mM Tris, pH 8.0, 150 mM KCl, 2 mM tris(2-carboxyethyl)phosphine [TCEP] and 10% glycerol. Fractions with pure protein were frozen at −80°C.

Calmodulin (from *Gallus gallus*) was expressed using a pET-15b vector (generous gift of Angus Nairn), and purified as described previously (*Putkey and Waxham, 1996*).

## Mass spectrometry

Samples for mass spectrometry were prepared by the addition of 80% acetonitrile and 250 ng trypsin to ~10 µM CaMKII. Reactions were carried out at 25°C for 2–3 hr then put on a Speed-Vac for an additional 2–3 hr to remove residual acetonitrile. Samples were analyzed using LC/MS (QB3/Chemistry Mass Spec Facility at UC Berkeley).

## Single molecule experiments

### Preparation of PEG coated glass surface for protein immobilization

Round glass coverslips (VWR) were cleaned by sonication in a 1:1 (vol/vol) mixture of isopropanol and water. They were then dried in nitrogen and further cleaned for 5 min in a Harrick Plasma PDC-32 G plasma cleaner and assembled immediately after plasma cleaning with an Attofluor cell chamber (Molecular Probes), adding 0.25 ml poly-L-lysine PEG with PLK-PEG-biotin (1 mg/ml; 500:3 ratio of the two solutions by volume, SuSoS AG). This procedure minimized noise from autofluorescent contaminants to a very low level. After 30 min, the samples were rinsed five times with phosphate-buffered saline (PBS). Streptavidin (Molecular Probes) was added to a final concentration of 0.1 mg/ml and incubated for 10 min. Excess streptavidin was rinsed with ten 1 ml rinses with PBS. Samples of streptavidin were incubated for an additional 30 min and then rinsed with five 5 ml portions of PBS.

## Mixing experiments

CaMKII is expressed with a C-terminal biotinylation sequence (Avitag) (*Howarth and Ting, 2008*), and after purification, labeled with biotin using the biotin ligase BirA. This reaction mixture contained 1 mM ATP, 10 µM biotin (dissolved in DMSO), 0.6 µM BirA ligase in a final volume of 0.5–1 ml. The reaction was carried out on ice for 1 hr and desalted into labeling buffer using a PD-25 column (GE Healthcare). Mass spectrometry indicates complete biotinylation.

Following biotinylation, CaMKII was labeled with Alexa fluor dyes. Purified CaMKII was desalted into a buffer containing 25 mM Tris, pH 8.0, 150 mM potassium chloride, 1 mM tris(2-carboxyethyl) phosphine (TCEP) and 10% glycerol just prior to labeling with Alexa Fluor $C_5$-maleimide dyes (488 and 594, Life Technologies). Alexa dye was resuspended in 25 mM Tris (pH 8.0) at a final concentration of 10–15 mM. Labeling was completed by addition of threefold to fivefold molar excess dye over CaMKII subunit concentration, and incubated for 2–3 hr at 25°C. Samples were desalted (1 or 2X depending on efficiency and amount of protein) using PD-25 columns (GE healthcare) into an imaging buffer (25 mM Tris, pH 8.0, 150 mM potassium chloride, 1 mM TCEP). Samples were concentrated using Amicon filters to a final subunit concentration of 6 µM. Dye incorporation was estimated using spectrophotometric analysis (Nanodrop, Thermo Scientific, DE). The absorbance of each dye at 280 nm ($A_{280}$) was estimated using free dye resuspended in buffer. The dye-labeled protein samples were then scanned and the $A_{280}$ was corrected for dye contribution. The concentration of protein and dye was calculated using the corresponding extinction coefficients and the ratio of protein labeled is estimated as: [dye]/[protein] (Alexa 488: $\varepsilon = 71,000$ cm$^{-1}$M$^{-1}$, Alexa 594: $\varepsilon = 71,000$ cm$^{-1}$M$^{-1}$, and extinction coefficients for CaMKII variants were estimated using an online protein calculator tool: http://web.expasy.org/protparam/).

CaMKII labeled with Alexa 488 (6 µM) was mixed with CaMKII labeled with Alexa 594 (6 µM) with or without ATP (250 µM), MgCl$_2$ (8 mM), calcium (500 µM) and calmodulin (6 µM). The final concentration of CaMKII in these mixtures is 4.8 µM. In the main text we refer to the stock concentration of CaMKII used to make the mixtures (6 µM). At each time point, 2 µl of the CaMKII mixture was diluted into 1000 µl of desalting buffer and spread onto the chamber housing the functionalized cover slip. CaMKII was incubated on the cover slip for 1 min and then the surface was washed 3X with 1 ml desalting buffer. These slides were then imaged using TIRF microscopy.

In experiments where the inhibitor, bosutinib, was added, 6 µM CaMKII (after labeling with Alexa fluor) was incubated with 50 µM bosutinib for 15 min at 25°C. Proteins were then mixed as above, except with less ATP (0 or 1 µM).

In the 3-color colocalization experiments, the pThr286 antibody (Pierce antibodies #A-20186, Thermo Scientific) was labeled with Alexa 647 prior to the experiment, following the protocol for the monoclonal antibody labeling kit (Life Technologies). At each time point in these experiments, antibody was added to the CaMKII mixture at 35-fold molar excess over CaMKII concentration and binding was carried out at 25°C for 3 min. The sample was then diluted 500-fold and spread onto the functionalized cover slip, as above.

For the preparation of phosphorylated CaMKIIα, the same procedure as above was followed, except that the His tag was not removed by Ulp1 cleavage, and a subtractive Ni-NTA purification step

was not done. This protein was then labeled with Alexa 594 as previously described. His-tagged CaMKII (6 μM) was activated at 25°C by addition of Ca$^{2+}$/CaM (40 μM/6 μM), ATP (250 μM) and MgCl$_2$ (8 mM) for 10 min. The reaction was quenched by addition of EDTA (50 mM) and EGTA (20 mM). This reaction volume was mixed with free Ni-NTA resin (1 ml), equilibrated with Buffer A (5% glycerol) and allowed to bind for 1–2 hr (4°C). After washing with Buffer A, bound CaMKII-His was eluted with 0.5 M imidazole (5% glycerol), desalted into imaging buffer using a PD-25 column and concentrated to at least 6 μM.

## Single-molecule TIRF microscopy

Single-particle fluorescence imaging was performed on a Nikon Ti-E/B (Tokyo, Japan) inverted microscope equipped with a Nikon 100x Apo TIRF 1.49 NA objective lens and a TIRF illuminator, Perfect Focus system, and a motorized stage (ASI MS-2000, Eugene OR). Static images were recorded with a Hamamatsu (Hamamatsu City, Japan) Orca-R2 interline charge coupled device (CCD) camera. The sample was illuminated with a 488 nm argon-ion laser (Spectra Physics 177g, Santa Clara, CA), 561 nm optically pumped solid state laser (Coherent Sapphire, Santa Clara, CA), 640 nm diode laser (Coherent Cube, Santa Clara, CA). Lasers were controlled using an acousto-optic tunable filter (AOTF) and aligned into a fiber launch custom built by Solamere (Salt Lake City, UT). A single-mode polarization maintaining fiber (Oz Optics, Ottawa, Canada) was connected to the TIRF illuminator. All optical filters were from Chroma (Bellows Falls, VT). Dichroics were 2-mm thick and mounted in metal cubes to preserve optical flatness: ZT488rdc, ZT561rdc, and ZT640rdc. Long-pass emission filters included: ET500lp, ET575lp, and ET660lp. Bandpass emission filters were located below the dichroic turret in a filter wheel (Sutter Lambda 10-3, Novato, CA): ET525/50m, ET600/50m, and ET700/75m. Multicolor acquisition was performed by computer-controlled change of illumination and filter sets. Multi-color TIRF imaging was performed at twelve different positions which were calculated from an initial reference point so as to capture images at a reasonable distance from one another. Images were acquired using Micro-Manager microscopy software. For all images, 2 by 2 binning was used as part of acquisition and 500 ms exposure time was used for most images.

## Colocalization calculation

Analysis of microscopy data was carried out using software written in Igor Pro ver. 6.22A (Wavemetrics, Oregon). The program enabled us to detect particles systematically and to quantify the amount of colocalization. Analysis was carried out in two overall steps: (i) localization of particles in micrographs and quantification of integrated fluorescence emission intensity from each identified spot and (ii) colocalization of detected particles in different color channels. Initial guesses for the location of particles were generated by locating sites of divergence in the gradient vector field of the image. Fitting an elliptical Gaussian to the intensity profile of individual particles refined the initial guesses. A cut-off in the circularity (evaluated as the ratio of the major and minor axis of the ellipse) enabled aggregates and particles with overlapping intensity distributions to be excluded. Colocalization, or the (number of colocalizing particles)/(number of total particles in reference channel), was evaluated by counting particles in separate channels with center positions located within a threshold distance (1.0–2.5 pixels). The colocalization number was normalized subsequently by the experimentally determined maximum value obtained from a positive control of doubly labeled CaMKII proteins. The channel with smaller total particle number was chosen systematically as the reference channel in all experiments. Error bars shown in the single-molecule experiments represent the standard deviation between multiple collected images.

## FRET solution experiments

A mutant of CaMKIIα was used for these experiments, in which all surface exposed cysteine residues (280, 289) were mutated to serine, and aspartate 335 was mutated to cysteine (D335C). The same labeling and mixing experiments for the single-molecule assay were done. Samples were incubated at 25°C or 37°C. At each time point, 25 μl from the mixed sample was removed and diluted to a final volume of 150 μl. An emission spectrum (500–700 nm) was acquired for each diluted sample excited at 490 nm using a Fluoromax-3 fluorometer (Horiba Scientific, Edison, NJ). Data were analyzed by calculating the FRET ratio (acceptor emission at 610 nm divided by donor emission at 515 nm). Error bars are calculated from the standard deviation between separate experiments on different days.

## Enzyme assays

Kinase activity was monitored using an ADP quest assay (*Charter et al., 2006*). Phosphorylated CaMKII (activated sample) was prepared as above. Activated and unactivated samples were diluted to 1 μM.

For the mixing experiments, these samples were mixed in a 3:1 ratio of unactivated:activated protein (15 µl unactivated + 5 µl activated). For controls, each sample was measured individually at these same concentrations, and brought to the same final volume by addition of buffer. The kinase assay was carried out in a 384-well plate format at 30°C in a 27.5 µl reaction volume. The reaction mixture contained the following components (listed at final concentration): 10 mM $MgCl_2$, 20 µM DTT, 0.3–0.5 mM peptide substrate syntide (PLARTLSVAGLPGKK), and Tris. For each reaction, 5 µl of the protein solution was mixed with 4 µl concentrated reaction mixture, and 1 µl of either water or $Ca^{2+}$/CaM (200 µM/4 µM) was added. The protocol for the ADP quest assay was followed for the remaining steps and reactions were initiated by the addition of 250 µM ATP to the mix. The increase in fluorescence was monitored at 590 nm (excitation at 530 nm) in a fluorescent microplate spectrophotometer (Synergy H4, Biotek) with sampling every 5 min. Slopes ($\Delta$fluorescence$_{590\ nm}$/time) were calculated from the early data points where an initial linear rate is observed.

## Molecular dynamics

The molecular dynamics trajectories were generated using the Gromacs 4.6.2 package (*Berendsen et al., 1995*; *Pronk et al., 2013*). For all the hub dimer molecular dynamics simulations, with the docked pThr containing peptide, Amber12 (*Case et al., 2012*) was used for ease of handling of modified backbone residues. The ff99SB-ILDN force field was used for all the calculations (*Lindorff-Larsen et al., 2010*). All simulations were carried out in aqueous medium using the TIP3P water model and appropriate counterions ($Na^+$ and $Cl^-$) were added to neutralize the net charges. After initial energy minimization, the systems were subjected to 30–100 ps of constant number, volume and temperature (NVT) equilibration, during which the system was heated to 300K. This was followed by a short equilibration at constant number, pressure and temperature (NPT, 20–100 ps). The equilibration steps were performed with harmonic positional restraints on the protein atoms. Finally, the production simulations were performed under NPT conditions, with the Berendsen and v-rescale thermostats in Amber12 and Gromacs 4.6.2 respectively, in the absence of positional restraints. Periodic boundary conditions were imposed, and particle-mesh Ewald summations were used for long-range electrostatics and the van der Waals cut-off is set at 1 nm. A time step of 2 fs was employed and the structures were stored every 2 ps. SHAKE and LINCS constraint algorithms were used with Amber12 and Gromacs 4.6.2, respectively, to fix covalent bonds.

## Modeling the docked peptide onto the hub assembly

We modeled a 16-residue stretch of the R3 element as an additional strand of the β sheet in the hub domain. We created the model by first docking a β strand taken from an arbitrary protein so that the last 9 residues of the strand were aligned roughly with the 8 residue strand formed by R3 element in the autoinhibited CaMKII holoenzyme (PDB code: 3SOA). This model does not have the strand aligned optimally with the rest of the β sheet, and to improve the model we searched the protein databank, using the PDBeFOLD server (*Krissinel and Henrick, 2004*), for structures that had β strands arranged in the same topology and with close spatial overlap to the three strands from the hub domain and the roughly modeled fourth β strand, yielding a match in the human core Snrnp domain (PDB code: 1D3B). We then created a new model in which the β sheet from the crystal structure of the hub domain was retained and the fourth strand was taken from Snrnp, with appropriate changes to the sequence.

Using this fourth strand as a template we modeled in the sequence of the CaMKII R3 element in more than one sequence register, and carried out a series of short molecular dynamics simulations (~100 ns each). In one of these simulations a conformational change that is similar to the transient fluctuations noted in the Appendix happened to occur, opening up access to the hub cavity. The sidechain of Met 307 from the R3 element was seen to approach Arg 403 closely in this simulation. We then changed the sequence register of the peptide so that Thr 305 occupied the position of the methionine sidechain, and we added a phosphate group to the threonine sidechain.

## Acknowledgements

We thank members of the Kuriyan and Groves labs, especially Jon Winger, for helpful discussion. We thank Jeff Iwig, Julie Zorn, Brian Kelch (UMass Worcester), and Charles Morgan (UCSF) for helpful comments on the manuscript, Tiago Barros for help with figure design, and Qi Wang for help with DLS experiments. We thank David King (HHMI, Mass Spectrometry) for generous assistance with synthesis of peptides. We thank Tony Iavarone (QB3) for mass spectrometry support and discussions, Stewart Loh and Anna Elleman for support, Giulio Superti-Furga for providing us with bosutinib, Joseph

Mougous (University of Washington) for the Hcp1 vector, and Alice Ting (M.I.T) for the BirA vector. We thank Eileen Jaffe for making us aware of the allosteric mechanism of porphobilinogen synthase.

## Additional information

### Competing interests

JK: Senior editor, *eLife*. The other authors declare that no competing interests exist.

### Funding

| Funder | Grant reference number | Author |
|---|---|---|
| Howard Hughes Medical Institute | | Jay T Groves, John Kuriyan |
| Jane Coffin Childs | | Margaret Stratton |
| National Institutes of Health | R01GM101277 | Howard Schulman |
| Human Frontier Science Program | | Moitrayee Bhattacharyya |

The funders had no role in study design, data collection and interpretation, or the decision to submit the work for publication.

### Author contributions

MS, I-HL, MB, Conception and design, Acquisition of data, Analysis and interpretation of data, Drafting or revising the article; SMC, HS, Analysis and interpretation of data, Drafting or revising the article; LHC, Conception and design, Drafting or revising the article; JTG, JK, Conception and design, Analysis and interpretation of data, Drafting or revising the article

## Additional files

### Major dataset

The following previously published datasets were used:

| Author(s) | Year | Dataset title | Dataset ID and/or URL | Database, license, and accessibility information |
|---|---|---|---|---|
| Hoelz A, Nairn AC, Kuriyan J | 2003 | Crystal Structure of Calciumcalmodulin-dependent Protein Kinase | 1HKX; http://www.rcsb.org/pdb/explore/explore.do?structureId=1HKX | Publicly available at the RCSB Protein Data Bank (http://www.rcsb.org/). |
| Chao LH, Stratton MM, Lee IH, Rosenberg OS, Levitz J, Mandell DJ, Kortemme T, Groves JT, Schulman H, Kuriyan J | 2011 | Full-length Human Camkii | 3SOA; http://www.rcsb.org/pdb/explore/explore.do?structureId=3SOA | Publicly available at the RCSB Protein Data Bank (http://www.rcsb.org/). |
| Rellos P, Pike ACW, Niesen FH, Salah E, Lee WH, Von Delft F, Knapp S | 2010 | Structure of the Oligomerisation Domain of Calcium- Calmodulin Dependent Protein Kinase II Gamma | 2UX0; http://www.rcsb.org/pdb/explore/explore.do?structureId=2UX0 | Publicly available at the RCSB Protein Data Bank (http://www.rcsb.org/). |
| Bullock TL, Clarkson WD, Kent HM, Stewart M | 1996 | Crystal Structure of Nuclear Transport Factor 2 (Ntf2) | 1OUN; http://www.rcsb.org/pdb/explore/explore.do?structureId=1OUN | Publicly available at the RCSB Protein Data Bank (http://www.rcsb.org/). |
| Mougous JD, Cuff ME, Raunser S, Shen A, Zhou M, Gifford CA, Goodman AL, Joachimiak G, Ordonez CL, Lory S, Walz T, Joachimiak A, Mekalanos JJ | 2006 | Structure of a Hemolysin-coregulated Protein From Pseudomonas Aeruginosa | 1Y12; http://www.rcsb.org/pdb/explore/explore.do?structureId=1Y12 | Publicly available at the RCSB Protein Data Bank (http://www.rcsb.org/). |

| Kambach C, Walke S, Young R, Avis JM, De La Fortelle E, Raker VA, Luhrmann R, Li J, Nagai K | 1999 | Crystal Structure of the D3B Subcomplex of the Human Core Snrnp Domain at 2.0a Resolution | 1D3B; http://www.rcsb.org/pdb/explore/explore.do?structureId=1D3B | Publicly available at the RCSB Protein Data Bank (http://www.rcsb.org/). |

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
