## [Decision Letter]

Thank you for sending your work entitled “Activation-triggered subunit exchange between CaMKII holoenzymes facilitates the spread of kinase activity” for consideration at *eLife*. Your article has been favorably evaluated by a Senior editor and 3 reviewers, one of whom, Wes Sundquis , is a member of our Board of Reviewing Editors.

The Reviewing editor and the other reviewers discussed their comments before we reached this decision, and the Reviewing editor has assembled the following comments to help you prepare a revised submission.

The authors present experiments and simulations showing that activated subunits from dodecameric CaMKII kinase assemblies can exchange into unactivated holoenzymes and thereby “spread” activation in a calcium-independent fashion. The authors make beautiful use of two- and three-color single molecule experiments (and complementary solution FRET experiments) and a series of mixed CaMKII constructs to demonstrate that subunit exchange and spreading occurs, and requires: 1) subunit activation, 2) dissociation of activated subunits from the dodecamer, 3) activities within the R3 regulatory and adjacent regulatory regions, including phosphorylation of Thr86 and Thr305/6, and 4) exchange of activated subunits into the previously inactive holoenzymes. An elegant mechanistic model for subunit exchange is proposed based upon structural and experimental constraints, and supported by molecular dynamics simulations.

The experiments are clean and elegant, the results remarkable, and the implications for CaMKII regulation and establishment of LTP are potentially far-reaching. The detailed proposal for activation-induced subunit exchange is still rather speculative, but is well thought out and logical and its inclusion enriches the paper. Unfortunately, the authors were unable perform the one very obvious test of their mechanistic proposal, mutation of the Arg-rich hub cavity, owing to technical problems. Nevertheless, there is much solid, important work here and it seems appropriate for the authors to propose a detailed model for subunit exchange and wait for subsequent studies to test the model (and its biological implications) more rigorously.

The paper is appropriate for *eLife* because the phenomenon reported is unprecedented, because it has physiological implications on long-term potentiation, and because the single molecule colocalization approaches are novel, especially the three color detection that includes a phosphorylation site. The work should therefore have a very broad appeal to the structural biology, biophysics, signaling, neurobiology, and single molecule imaging communities.

Experimental issues:

1) Some of the assays used to demonstrate subunit exchange use wt forms of CaMKII (not pseudophosphorylated at T286). The authors deal with the important issue of whether activation conditions lead to aggregation of CaMKII into large clusters, as previously observed by others. They claim “no evidence” for such aggregation, but test this only for the CaMKII T286D form. Because their mixing experiments use other forms, they need to provide evidence regarding aggregation of these forms.

2) The authors should explain why they did not use a kinase defective mutant to test the requirement of the kinase activity. Instead, they used the unusual approach of lowering ATP concentration to 1 μM (why not zero?) and adding a kinase inhibitor. Using a kinase dead mutant would seem to be more direct and convincing. Was this approach used because a kinase mutant is not available or because such a study was performed but subunit exchange was still observed?

3) It is unsatisfying that the authors did not attempt to quantify the apparent stoichiometry of protein complexes through photobleaching analysis. They will get a distribution, of course, but they should be able to determine how many of their fluorescent spots are due to potential dimers because if dimers are indeed ejected upon activation, there will be more spots with fewer photobleaching steps. It will also be good to perform an experiment with a single population of wild-type kinase all labeled with a single dye and measure the distribution of photobleaching step counts before and after activation. The current single molecule intensity distribution measurements are not precise enough to address these points.

Conceptual/discussion issues:

1) The authors consistently refer to subunit exchange from an active holoenzyme into an inactive one. As we understand the mixing experiments, the exchange could also be from inactive form into an active form. If this is correct, the wording should be changed. This issue is important because it relates to one of the physiological roles of CaMKII that is alluded to: the role of subunit exchange in memory maintenance. According to ideas cited, CaMKII bound to the NMDA channel may be a molecular memory at synapses. The work of Colbran (2011) shows the importance of autophosphorylated CaMKII for binding to the PSD. One can thus assume that the bound CaMKII is autophosphorylated and active. If such a holoenzyme were replaced during protein turnover by an unphosphorylated holoenzyme, information would be lost. On the other hand, if as postulated in previous models (Lisman), turnover occurred by subunit exchange, the key step in protein turnover would be insertion of unphosphorylated (new) subunits into a previously activated holoenzyme. The new subunit would become phosphorylated by well established intra-holoenzyme autophosphorylation. In this way, any loss of phosphorylated subunits from this holoenzyme would be compensated. These ideas are not clearly presented in the Discussion. The Discussion is further muddled by allusion to a completely different phenomenon, the spread to CaMKII activation throughout the dendrite after local stimulation. This is a little studied phenomenon of questionable physiological significance. It is worthy of mention but needs to be described in a separate paragraph and not mixed in with the discussion of the potential importance of subunit exchange for memory maintenance.

2) The authors should make it explicit that 305/306 regulation of subunit exchange might be just modulatory. There is no evidence that it is necessary.

3) For much of the data presented, one could have argued that the complex stability decreases with activation, leading to disassembly and reassembly resulting in mixing. Therefore, the data presented in Figure 5 is important but the readers have to read through other details, for example whether certain linkers and artificial multimerization domain are important for the exchange. The authors should consider moving Figure 5 up as much as possible to establish this most important aspect of the work.

4) The authors should also consider that some of the readers will be interested in how far this activation can propagate, that is what is the amplification factor? Will a single activated complex induced activation of one more, 10 more, or 100 more, for example. Their data as a function of protein concentration showing a steep drop in colocalization at low concentrations seems suitable for this type of modeling.

The spread of activation could indeed be mediated by the insertion of an active subunit into an inactive one. However, measurements done under physiological conditions (Yasuda Lab) show that bulk CaMKII is rapidly dephosphorylated. Thus, under these conditions, insertion of an active subunit into an inactive holoenzyme would not lead to “gain”. Indeed, such gain would be undesirable because it would propagate the biochemical signal that triggers synaptic modification to inactive synapses. Yasuda's work on optical detection of activated CaMKII shows that it remains localized to the activated spine.

---

## [Author Response]

Experimental issues:

*1) Some of the assays used to demonstrate subunit exchange use wt forms of CaMKII (not pseudophosphorylated at T286). The authors deal with the important issue of whether activation conditions lead to aggregation of CaMKII into large clusters, as previously observed by others. They claim “no evidence” for such aggregation, but test this only for the CaMKII T286D form. Because their mixing experiments use other forms, they need to provide evidence regarding aggregation of these forms*.

The reviewers are correct to point out that it is important to analyze the state of aggregation of CaMKII for both wild-type enzyme activated by Ca^2+^/CaM and the constitutively active T286D mutant. Regarding the observation by others of large scale aggregates in the presence of Ca^2+^/CaM, we note that our earlier analysis of the *C. elegans* holoenzyme by small angle X-ray scattering (SAXS) found no evidence for aggregation in the presence of Ca^2+^/CaM, with the only structural effect of Ca^2+^/CaM being an increase in the radius of gyration of the holoenzyme (see Figure 7 of [58]). We suspect that the aggregation observed by others may be a consequence of differences in the solution conditions, particularly the pH, but we have not investigated this further.

We now present dynamic light scattering analysis for the wild-type human enzyme activated by Ca^2+^/CaM to augment the data for the T286D mutant that had been presented in the original manuscript (Figure 2—figure supplement 1). These data demonstrate that there is no significant time dependent aggregation for either sample. It is interesting to note that the hydrodynamic radius derived from the light scattering data for the Ca^2+^/CaM-activated sample (∼30 nm) is larger than that for the T286D mutant (∼11 nm). We interpret the apparent increase in the hydrodynamic radius in the presence of Ca^2+^/CaM to mean that the binding of CaM converts the holoenzyme to a more extended form than seen for the T286D mutant. Regardless of the origin of the increased apparent hydrodynamic radius of the Ca^2+^/CaM complex, there is no observable increase in the hydrodynamic radius with time for either sample. Note also that the T286D mutant exhibits a robust degree of subunit exchange in the absence of Ca^2+^/CaM, comparable to the exchange kinetics seen for the wild-type enzyme in the presence of Ca^2+^/CaM. Thus, the differences in the apparent hydrodynamic radii of the two species is not a contributing factor to the exchange process.

We also now present intensity analysis for the single-molecule experiments for the T286D mutant as well as for the wild-type enzyme with Ca^2+^/CaM (the original manuscript presented data only for the wild-type enzyme; Figure 2—figure supplement 3). These data make clear that there is no time-dependent increase in the intensities of the samples, consistent with the absence of aggregation in the samples analyzed by the single-molecule method.

*2) The authors should explain why they did not use a kinase defective mutant to test the requirement of the kinase activity. Instead, they used the unusual approach of lowering ATP concentration to 1 μM (why not zero?) and adding a kinase inhibitor. Using a kinase dead mutant would seem to be more direct and convincing. Was this approach used because a kinase mutant is not available or because such a study was performed but subunit exchange was still observed*?

A kinase inhibitor was used in place of a kinase-dead mutant in order to compare exchange rates for the same protein preparation directly. We now present a FRET experiment to test exchange in the presence of bosutinib and no ATP, which was a good suggestion by the reviewers (Figure 4—figure supplement 1). The data show clearly that in the presence of Ca^2+^/CaM and a kinase inhibitor with no ATP added, the exchange is suppressed to the level of unactivated CaMKII.

*3) It is unsatisfying that the authors did not attempt to quantify the apparent stoichiometry of protein complexes through photobleaching analysis. They will get a distribution, of course, but they should be able to determine how many of their fluorescent spots are due to potential dimers because if dimers are indeed ejected upon activation, there will be more spots with fewer photobleaching steps. It will also be good to perform an experiment with a single population of wild type kinase all labeled with a single dye and measure the distribution of photobleaching step counts before and after activation. The current single molecule intensity distribution measurements are not precise enough to address these points*.

The reviewers are correct in saying that our current intensity measurements are insufficient to derive stoichiometric information regarding the exchange process. For the single molecule studies shown in the original manuscript, the labeling efficiency was well below 100%, which would affect any type of stoichiometric analysis. Additionally, if each subunit of a CaMKII dodecamer was labeled with one dye moiety, this would create more photobleaching steps than can accurately be measured. The numerous cysteine residues in the wild-type CaMKII sequence also introduce an uncertainty in the labeling distribution. We have performed photobleaching experiments using our current setup (see sample traces below [Figure 11], which are provided here but will not be included in the final manuscript).Author response image 1.Author response image 1: Sample traces

The information obtained from these traces does not yield significant information regarding stoichiometries of the unit of exchange. Absolute counting of photobleaching steps is not a straightforward way to obtain the true stoichiometry of multimeric protein samples and models correcting for submaximal labeling of proteins with chromophores, non-linear detection of signals with different brightness and photophysical effects must be employed and validated with careful control experiments. Follow-up experiments in the future will certainly be geared toward this important question.

*Conceptual/discussion issues*:

*1) The authors consistently refer to subunit exchange from an active holoenzyme into an inactive one. As we understand the mixing experiments, the exchange could also be from inactive form into an active form. If this is correct, the wording should be changed. This issue is important because it relates to one of the physiological roles of CaMKII that is alluded to: the role of subunit exchange in memory maintenance. According to ideas cited, CaMKII bound to the NMDA channel may be a molecular memory at synapses. The work of Colbran (2011) shows the importance of autophosphorylated CaMKII for binding to the PSD. One can thus assume that the bound CaMKII is autophosphorylated and active. If such a holoenzyme were replaced during protein turnover by an unphosphorylated holoenzyme, information would be lost. On the other hand, if as postulated in previous models (Lisman), turnover occurred by subunit exchange, the key step in protein turnover would be insertion of unphosphorylated (new) subunits into a previously activated holoenzyme. The new subunit would become phosphorylated by well established intra-holoenzyme autophosphorylation. In this way, any loss of phosphorylated subunits from this holoenzyme would be compensated. These ideas are not clearly presented in the Discussion. The Discussion is further muddled by allusion to a completely different phenomenon, the spread to CaMKII activation throughout the dendrite after local stimulation. This is a little studied phenomenon of questionable physiological significance. It is worthy of mention but needs to be described in a separate paragraph and not mixed in with the discussion of the potential importance of subunit exchange for memory maintenance*.

The reviewers are correct to point out that our current understanding of the exchange process does not clearly delineate directionality of subunit exchange between dodecamers. We have reworded this where appropriate. Additionally, we have reorganized the Discussion to separate the idea of CaMKII spreading throughout the dendritic spine after activation from the Discussion of the effects of exchange on CaMKII bound to the NMDA receptor.

*2) The authors should make it explicit that 305/306 regulation of subunit exchange might be just modulatory. There is no evidence that it is necessary*.

This wording has been corrected.

*3) For much of the data presented, one could have argued that the complex stability decreases with activation, leading to disassembly and reassembly resulting in mixing. Therefore, the data presented in*
Figure 5
*is important but the readers have to read through other details, for example whether certain linkers and artificial multimerization domain are important for the exchange. The authors should consider moving*
Figure 5
*up as much as possible to establish this most important aspect of the work*.

We appreciate this comment and had attempted to reorganize the manuscript in this way. A large number of experiments are discussed in the paper, and we found that the rearrangement made it harder to keep the discussion coherent. We have, therefore, retained the original flow of the paper.

*4) The authors should also consider that some of the readers will be interested in how far this activation can propagate, that is what is the amplification factor? Will a single activated complex induced activation of one more, 10 more, or 100 more, for example. Their data as a function of protein concentration showing a steep drop in colocalization at low concentrations seems suitable for this type of modeling*.

*The spread of activation could indeed be mediated by the insertion of an active subunit into an inactive one. However, measurements done under physiological conditions (Yasuda Lab) show that bulk CaMKII is rapidly dephosphorylated. Thus, under these conditions, insertion of an active subunit into an inactive holoenzyme would not lead to “gain”. Indeed, such gain would be undesirable because it would propagate the biochemical signal that triggers synaptic modification to inactive synapses. Yasuda's work on optical detection of activated CaMKII shows that it remains localized to the activated spine*.

From our current set of data, we cannot accurately estimate the effect of activation spread in terms of total propagation. We are planning future experiments and extensive modeling to try to answer this difficult question. However, this will most likely require devising in vivo experimentation.